# Illusory adversarial attacks on sequential decision-makers and countermeasures

## Abstract

Autonomous decision-making agents deployed in the real world need to be robust against possible adversarial attacks on sensory inputs. Existing work on adversarial attacks focuses on the notion of *perceptual invariance* popular in computer vision. We observe that such attacks can often be detected by victim agents, since they result in *action-observation sequences* that are not consistent with the dynamics of the environment. Furthermore, real-world agents, such as physical robots, commonly operate under human supervisors who are not susceptible to such attacks. We propose to instead focus on attacks that are *statistically undetectable*. Specifically, we propose *illusory* attacks, a novel class of adversarial attack that is consistent with the environment dynamics. We introduce a novel algorithm that can learn illusory attacks end-to-end. We empirically verify that our algorithm generates attacks that, in contrast to current methods, are undetectable to both AI agents with an environment dynamics model, as well as to humans. Furthermore, we show that *existing* robustification approaches are relatively ineffective against illusory attacks. Our findings highlight the need to ensure that real-world AI, and human-AI, systems are designed to make it difficult to corrupt sensory observations in ways that are consistent with the environment dynamics.

## 1 Introduction

Deep reinforcement learning algorithms (Mnih et al., 2015; Schulman et al., 2017; Haarnoja et al., 2018; Salimans et al., 2017, DQN, PPO, SAC, ES) have found applications across a number of sequential decision-making problems, ranging from simulated and real-world robotics (Todorov et al., 2012; Andrychowicz et al., 2020) to arcade games (Mnih et al., 2015). It has recently been found, however, that deep neural network control policies conditioning on high-dimensional sensory input are prone to adversarial attacks on the input observations, which poses threats to security and safety-critical applications (Kos & Song, 2017; Huang et al., 2017) and thus motivates research into robust learning algorithms (Zhang et al., 2020).

Existing frameworks of attacks on sequential decision-makers are largely inspired by pioneering work on perceptually invariant attacks in supervised computer vision settings (Ilahi et al., 2021). Unlike supervised settings, however, sequential decision-making settings involve temporally-extended environment interactions which give rise to temporally-correlated sequences of observations. In this paper, we argue that their failure to take temporal consistency considerations into account renders existing observation-space adversarial attacks ineffective in many settings of practical interest.

AI agents often have access to an approximate or exact *world model* (Sutton, 2022; Ha & Schmidhuber, 2018). In addition, humans have the ability to perform "intuitive physics" (Hamrick et al., 2016), using robust but qualitative internal models of the world (Battaglia et al., 2013). This makes it possible to use one's understanding of the world to detect a large range of existing adversarial attacks, by spotting inconsistencies in observation sequences. Existing observation-space adversarial attacks (Ilahi et al., 2021; Chen et al., 2019; Qiaoben et al., 2021; Sun et al., 2020) ignore these facts. As a result, these attacks produce observation trajectories that are inconsistent with the dynamics of the unattacked environment.

The consequences of this are twofold. First, state-of-the-art adversarial attacks can be trivially detected by victim agents with access to even low-accuracy world models, i.e., models of the environment dynamics. Second, in cases where AI agents are supervised by humans, humans ~~may also~~ can

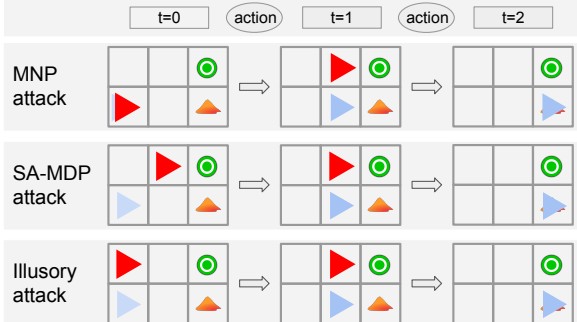

Figure 1: Illustration of different classes of adversarial attacks on a 6-cell Gridworld environment. Victim agents need to reach the green target as quickly as possible without traversing the orange lava. The adversary replaces the original victim observation (blue triangle) with an adversarial observation (red triangle). Note that in all scenarios, the victim ends up in the lava, upon which the episode terminates. However, the observations under the MNP and SA-MDP attacks (see Section 3) are not consistent with the *forward* actions taken by the agent, i.e. the red arrow jumps between cells (top row), respectively incorrectly stays in the same position (middle row). In contrast, the observations under the proposed illusory attack (bottom row) are consistent with the environment dynamics.

be able to detect these adversarial attacks. In other security contexts such as computer networks, real-world adversarial attacks typically attempt to evade detection in an effort to avoid triggering security escalations, and this must be taken into account by the defender. While this insight has been exploited in the cybersecurity community (Provos, 2001; Claffy & Dainotti, 2012; Cazorla et al., 2018), *undetectable* adversarial attacks on sequential decision-makers and their defences have not yet been systematically explored in the AI community.

In this paper, we introduce *illusory attacks*, a novel class of adversarial attacks on sequential decision-makers that result in observation space attacks that are consistent with environment dynamics. We show that illusory attacks can succeed where existing attacks do not, and, in particular, can successfully fool humans. Illusory attacks therefore pose a specific safety and security threat to human–AI interaction and human-in-the-loop settings (Schirner et al., 2013; Zerilli et al., 2019) as they ~~may~~ can be undetectable even to attentive human supervisors.

Illusory attacks seek to remain undetected and must hence attack the victim by replacing its perceived reality by an internally coherent alternative one. We show that *perfect* (statistically undetectable) illusory attacks exist in a variety of environments. We then present the $\mathcal{W}$-illusory attack framework (see Figure 1 for illustration), which introduces a *world-model consistency* optimisation objective that encourages the resultant victim's action-observation histories to be consistent with the environment dynamics. We show that $\mathcal{W}$-illusory attacks can be efficiently learnt, and are undetectable by humans, unlike MNP (Kumar et al., 2021) or SA-MDP attacks (Zhang et al., 2021).

We empirically demonstrate that existing victim robustification methods are largely ineffective against illusory attacks. This leads us to suggest that the existence of *reality feedback*, i.e., observation channels that are hardened against adversarial interference, ~~may~~ can play a decisive role in certifying the safety of real-world AI, and human–AI, systems.

Our work makes the following contributions:

- We formalise *perfect illusory attacks*, a novel framework for undetectable adversarial attacks (see Section 4.2). We give examples in common benchmark environments in Section 5.3.
- We introduce $\mathcal{W}$-illusory attacks, a computationally feasible learning algorithm for adversarial attacks that generate victim action-observation sequences that are consistent with the unperturbed environment dynamics (see Section 4.5).
- We show that, compared to state-of-the-art adversarial attacks, $\mathcal{W}$-illusory attacks are significantly less likely to be detected by AI agents (Section 5.2), as well as by humans (Section 5.4).
- We demonstrate that victim robustification against $\mathcal{W}$-illusory attacks is challenging unless the environment admits reality feedback (see Section 5.5).

## 2 RELATED WORK

**Adversarial attacks** literature originates in non-sequential decision-making applications such as image classification (Szegedy et al., 2013), where the goal is to find perturbations $\delta$ for a given classifier $f$ such that $f$ yields different predictions for $x$ and $x + \delta$, despite the difference between $x$ and $x + \delta$ being imperceptible to humans. To enforce the imperceptibility requirement, such works use simple minimum-norm perturbations constraints as a proxy (Goodfellow et al., 2014).

This line of work has been extended to adversarial attacks on **sequential decision-making agents** (Oikarinen et al., 2021b; Kumar et al., 2021; Cohen et al., 2019; Everett et al., 2021; Moosavi-Dezfooli et al., 2017; Chaubey et al., 2020; Wu et al., 2021; Ilahi et al., 2021; Chen et al., 2019; Qiaoben et al., 2021; Sun et al., 2020; Zhang et al., 2020), largely building upon the minimum-norm perturbation (MNP) framework. In the MNP framework, the adversary can modify the victim's observations up to a fixed step- or episode-wise perturbation budget. While many adversarial attack frameworks require white-box access to the victim's policy, Zhang et al. (2020) and Sun et al. (2021) use reinforcement learning to learn adversarial policies, thus require only black-box access to the victim's policy. Assuming a different setting, Hussenot et al. (2019) introduce a class of adversaries for which a unique mask is precomputed and added to the agents' observation at every time step. Our framework differs from these previous works both in that it takes into account the temporal correlation of sequences of observations, and in that it focuses on learning adversarial attacks that are undetectable.

Work towards **robust agents in sequential decision environments** uses randomized smoothing (Kumar et al., 2021; Wu et al., 2021), test-time hardening by computing confidence bounds (Everett et al., 2021), training with adversarial loss functions (Oikarinen et al., 2021a), and co-training with adversarial agents (Zhang et al., 2021). We compare against, and build upon, this line of work.

Another extensive body of work focuses on **detecting adversarial attacks**. Lin et al. (2017) develop an action-conditioned frame module that allows agents to detect adversarial attacks by comparing both the module's action distribution with the realised action distribution. Tekgul et al. (2021) detect adversaries by evaluating the feasibility of past action sequences. Compared to these works, our proposed methods hinge on detectability based on stronger notions of statistical indistinguishability. Further, it focuses on *learning* undetectable attacks, rather than detecting adversarial attacks.

There exists only a limited body of literature on undetectable (or **"stealthy"**) adversarial attacks (Li et al., 2019; Sun et al., 2020) on sequential-decision-making agents, none of which achieves statistical indistinguishability, nor takes into account temporally-extended observation dependencies; Huang & Zhu (2019) consider stealthy attacks on reward signals used for RL. In cybersecurity, similar ideas have been explored for sensor attacks on linear control systems (Mo & Sinopoli, 2010; Pasqualetti et al., 2015).

## 3 BACKGROUND

**MDP and POMDP.** A Markov decision process (Bellman, 1958, MDP) is a tuple $\langle \mathcal{S}, \mathcal{A}, p, r, \gamma \rangle$ where $\mathcal{S}$ is a discrete or continuous state space, $\mathcal{A}$ is a discrete or continuous action space, $p : \mathcal{S} \times \mathcal{A} \times \mathcal{S} \to [0, 1]$ is the state transition probability, $r : \mathcal{S} \times \mathcal{A} \times \mathbb{R} \to [0, 1]$ is a probabilistic reward function, and $\gamma \in [0, 1]$ is a scalar discount factor. A Markov decision process proceeds by sampling an initial state $s_0 \sim p(\emptyset) \in \mathcal{S}$, upon which the agent ~~may take~~ takes an action $a_0 \sim \pi(s_0) \in \mathcal{A}$ triggering a state transition $s_1 \sim p(s_0, a_1)$ and returning a reward $r_0 \sim r(s_1, a_0, s_0)$. Here, $\pi : \mathcal{S} \times \mathcal{A} \to [0, 1]$ is a stochastic agent policy.

A partially-observable MDP (Åström, 1965, POMDP) is a *latent* MDP $\langle \mathcal{S}, \mathcal{A}, p, r, \gamma \rangle$, together with a stochastic observation function $\mathcal{Z} : \mathcal{S} \times \mathcal{O} \to [0, 1]$. In a POMDP, the agent cannot observe states $s_t$ directly, but has to act based on observations $z_t \sim \mathcal{Z}(s_t)$.

We further define a ~~state-observation~~ action-observation trajectory of length $n$ as ~~$\tau = (z_0, a_0, \ldots, z_n)$~~ $\tau = (z_0, a_0, r_0, \ldots, z_n)$, and the distribution over trajectories generated by a policy $\pi$ acting in a (PO)MDP $\mathcal{E}$ as $\mathbb{P}_{\mathcal{E},\pi}(\tau)$. We assume that agents do not have access to the reward signal at test-time. We denote that two distributions $\mathbb{P}_{\mathcal{E},\pi}(\tau), \mathbb{P}'_{\mathcal{E},\pi}(\tau)$ are identical *up to reward* by writing ~~$\mathbb{P}_{\mathcal{E},\pi}(\tau) = \mathbb{P}'_{\mathcal{E},\pi}(\tau)$~~ $\mathbb{P}_{\mathcal{E},\pi}(\tau) \overset{\neg r}{=} \mathbb{P}'_{\mathcal{E},\pi}(\tau)$.

**Observation-space adversarial attacks on sequential decision making agents.** Observation-space adversarial attacks (Kumar et al., 2021) consider the scenario where an *adversary* can manipulate the observation given to a *victim* agent at test-time, thus introducing partial observability to the victim's test-time environment. The victim's observation at time step $t$ is then given as $\tilde{z}_t = z_t + \epsilon_t$ instead of $z_t$, where $\epsilon_t \in \mathbb{R}$ is the perturbation introduced by the adversary. When modeling the adversarial perturbations as generated by a state-conditioned adversarial agent with policy $\pi_{adv}$, we can write $\tilde{z}_t = \pi_{adv}(s_t)$. In many scenarios, the size of the perturbation is bounded by a budget $B > 0$; for simplicity, we consider zero-sum adversarial attacks, where the adversary minimizes the expected return of the victim. This yields the following definition of an *optimal state-conditioned observation-space adversary*:

$$\pi_{\text{adv}}^* = \arg\min_{\pi_{\text{adv}}} \mathbb{E}_{\pi_v}\left[\sum_{t=0}^{T-1} r_t\right] \qquad \text{s. t.} \qquad \sqrt{\sum_{t=0}^{T-1} \|\epsilon_t\|_2^2} \leq B, \tag{1}$$

where $a_t \sim \pi_v\big(\cdot | \tilde{z}_t\big), \tilde{z}_t \sim \pi_{adv}(s_t), s_{t+1} \sim p(\cdot | s_t, a_t)$.

This concept can be formalized as a state-adversary MDP SA-MDP (Zhang et al., 2020), which is defined as a tuple of the victim MDP $\langle \mathcal{S}, \mathcal{A}, p, r, \gamma \rangle$ together with an adversary $\nu : \mathcal{S} \to \mathcal{P}(S)$ and a mapping $\mathcal{B} : \mathcal{S} \to \mathcal{S}$, limiting supp $\nu(\cdot|s) \in \mathcal{B}(s)$.

**Robustifying against adversarial attacks with co-training.** The optimal policy $\pi_v^*$ in an SA-MDP thereby defines the optimal robust policy of a victim under the presence of an optimal adversary $\nu^*$. To find such optimally robust victim policies, Zhang et al. (2021, ATLA) introduce a *co-training* approach to SA-MDPs, where victim and adversary are trained in turn while keeping the other's policy fixed.

## 4 METHODS

In this paper, we are concerned with finding adversarial attack policies $\pi_a$ that are *undetectable*. Our setting differs from traditional observation-space attacks (see Section 3) in three ways. First, we assume that the victim has access to a *world model* $m_v$. Second, ~~we assume that perturbation budgets may be unconstrained~~ we consider unconstrained perturbation budgets. Third, we assume that the victim has a costly contingency option which it can safely execute once it has detected the presence of a test-time adversary.

While seemingly unusual, we argue that the above three assumptions are realistic in many settings: Often victims can learn accurate world models from unperturbed train-time samples, or rely on the experience of a human supervisor for anomaly detection. In addition, in some settings the constraint on small perturbations seems arbitrary, e.g. when an attacker is replacing the entire video feed of a surveillance camera. Lastly, contingency options for AI agents are a common feature of real-world systems, such as e.g. self-driving cars that can hand-over to a human controller, or security level escalation procedures in cyber-physical systems (Züren, 2021).

### 4.1 DETECTABILITY OF ADVERSARIAL ATTACKS

An adversarial attack can be detected if the *attacked* MDP $\mathcal{E}'$, which includes the adversary with $\pi_{adv}$, can be statistically distinguished from the train-time MDP $\mathcal{E}$.

**Definition 4.1** (Statistical indistinguishability). Let $\mathcal{E} := \langle \mathcal{S}, \mathcal{A}, p, r, \gamma \rangle$ be a MDP with horizon T. Let $\mathcal{P}$ be a partially-observable discrete-time stochastic control process, with observation space $\mathcal{Z} := \mathcal{S}$, action space $\mathcal{A}$, and horizon T. Let $\pi : \mathcal{S} \times \mathcal{A} \times (\mathcal{S} \times \mathcal{A})^{T-1} \to [0, 1]$ be a sampling policy that ~~may~~ possibly conditions on the whole action-observation history. Then $\mathcal{P}$ is *statistically indistinguishable* from $\mathcal{E}$ under $\pi$ if and only if $\mathbb{P}_{\pi,\mathcal{E}} = \mathbb{P}_{\pi,\mathcal{E}'}$ $\mathbb{P}_{\mathcal{E},\pi} \overset{\urcorner r}{=} \mathbb{P}_{\mathcal{P},\pi}$.

### 4.2 UNDETECTABLE ADVERSARIAL ATTACKS

We define *perfect illusory* attacks, i.e. adversarial attacks that are undetectable, as those where $\mathcal{E}$ and $\mathcal{E}'$ are statistically indistinguishable. Importantly, since in our setting reward signals cannot be

observed at test-time, indistinguishability need only hold for observation transitions. For simplicity, we only consider zero-sum adversarial attacks, i.e. those that try to minimise the expected return of the victim.

**Definition 4.2** (Perfect illusory attack). A perfect illusory attack on an environment $\mathcal{E}$ for a given victim policy $\pi_v$ is an adversarial policy $\pi_a$, such that $\mathcal{E}$ and $\mathcal{E}'$ are statistically indistinguishable under $\pi_v$, where $\mathcal{E}'$ is the environment included by $\pi_a$ attacking $\mathcal{E}$.

Note that the definition of perfect illusory attacks makes no reference to the adversary's performance (victim reward). We prove that non-trivial (i.e. those actually changing the observation) perfect illusory attacks do exist in some, but not all, victim policy/environment pairs (see Appendix 7.1). We provide further examples of perfect illusory attacks in Section 5.

**Definition 4.3** (Optimal illusory attack). An illusory attack with $\pi_{adv}$ on $(\mathcal{E}, \pi_v)$ is optimal if and only if it is the perfect illusory attack with the highest expected adversarial return, i.e.

$$\pi^*_{adv} = \arg\min_{\pi_{adv}} \mathbb{E}_{\tau \sim \mathbb{P}(\mathcal{E}', \pi_v)} \left[ \sum_{t=0}^{T-1} r_t \right],$$

$$\text{s.t. } \underline{\mathbb{P}_{\mathcal{E},\pi_v} = \mathbb{P}_{\mathcal{E}',\pi_v}} \mathbb{P}_{\mathcal{E},\pi_v} \overset{\neg r}{=} \mathbb{P}_{\mathcal{E}',\pi_v} \tag{2}$$

, where $r_t$ is the victim reward.

As the constraints in Equation 2 are difficult to optimise exactly, we instead consider a relaxed weighted objective

$$\pi^*_{adv} = \arg\min_{\pi_{adv}} \mathbb{E}_{\tau \sim \mathbb{P}(\mathcal{E}', \pi_v)} \left[ \sum_{t=0}^{\underline{T}T-1} r_t \right] + \lambda \mathcal{D}\left( \mathbb{P}_{\mathcal{E},\pi_v}, \mathbb{P}_{\mathcal{E}',\pi_v} \right), \tag{3}$$

where $\lambda$ is a hyper-parameter that determines the weighing of the two objectives, and $\mathcal{D}$ is some distance measure between distributions, such as the KL-divergence.

**Adversaries that trade off between remaining undetected and minimizing the victim reward.** In most environments, an adversary will need to trade-off its expected adversarial return with the desire to avoid detection. In practice, ~~illusory attacks may not have to be perfect in order to be effective~~ even non-perfect illusory attacks can be effective, for example when victims only have access to only $n$ test-time samples from $\mathcal{E}'$. In fact, $n$ can usually be actively upper-bounded by the adversary's preferences to engage or not. In addition, the victim's world model ~~may not be exact~~ can be imperfect, particularly if learnt, or if relying on human experience. We define illusory attacks under such relaxations in Appendix 7.2.

### 4.3 WORLD MODEL-CONSISTENT ADVERSARIAL ATTACKS.

Analytically optimising the distance $\mathcal{D}\left( \mathbb{P}_{\mathcal{E},\pi_v}, \mathbb{P}_{\mathcal{E}',\pi_v} \right)$ is usually intractable in all but the smallest environments. To arrive at a tractable optimisation objective, we instead consider a two-step test that allows to determine whether $\hat{\mathbb{P}}_{\mathcal{E},\pi_v} \overset{\neg r}{=} \hat{\mathbb{P}}_{\mathcal{E}',\pi_v}$.

**Theorem 4.4** (Testing MDP equivalence). *To determine whether a discrete-time stochastic control process $\mathcal{M}$ is equivalent to a given Markov process $\mathcal{E}$, it is sufficient to test both whether $\mathcal{M}$ has the Markov property, and whether $\mathcal{M}$'s transition probabilities match those of $\mathcal{E}$ (Shi et al., 2020).*

We further assume from now that the victim has access to an exact, or approximate, world model $m_v(z'|z, a)$ of $\mathcal{E}$, which estimates the probability of the next observations given the current observation and action taken in $\mathcal{E}$. We also assume the victim has an estimate $\hat{\mathbb{P}}(z'|z, a)$ of the same quantity but in $\mathcal{E}'$.

We now define $\mathcal{W}$-illusory attacks, a relaxation of optimal illusory attacks (see Definition 4.3), for which the victim's world model estimate matches the estimate of the transition probabilities.

**Definition 4.5** ($\mathcal{W}$-illusory attacks)**.** A $\mathcal{W}$-illusory attack on $\mathcal{E}$ is an adversarial attack $\pi_a$ that is consistent with the victim's model of the observation-transition probabilities $m_v$. Formally,

$$\pi_{adv}^* = \arg\min_{\pi_{adv}} \mathbb{E}_{\pi_v} \left[ \sum_{t=0}^{T-1} r_t \right],$$

$$\text{s.t. } m_v(\cdot | z_t, a_t) = \hat{\mathbb{P}}(\cdot | z_t, a_t) \ \forall z_t, a_t \sim \langle \mathcal{E}', \pi \rangle \tag{4}$$

We note that world-model consistency alone *does not imply* the Markov property, as $\hat{\mathbb{P}}$ does not account for long term correlations. So importantly, $\mathcal{W}$-illusory attacks can in general *change the distribution* of trajectories.

### 4.4 Learning world model consistent adversarial attacks.

We now define the Illusory-MDP ($\mathcal{I}$-MDP) used to learn illusory adversarial attacks. This is the MDP the adversary trains on.

**Definition 4.6** (Illusory MDP ($\mathcal{I}$-MDP))**.** Given a MDP $\mathcal{E} := \langle \mathcal{S}, \mathcal{A}, p, r, \gamma \rangle$ with episode horizon $T$ and a victim policy $\pi$, an *Illusory MDP ($\mathcal{I}$-MDP)* $\mathcal{E}_\mathcal{I} := \mathcal{I}(\mathcal{E}, \pi)$ consists of a tuple $\left\langle \tilde{\mathcal{S}}, \tilde{\mathcal{A}}, \tilde{p}, \tilde{r}, \gamma \right\rangle$ where $\tilde{\mathcal{A}} := \mathcal{S}, \tilde{\mathcal{S}} := \mathcal{S} \times \left( \tilde{\mathcal{A}} \times \mathcal{A} \right)^{T-1}$,

$$\tilde{p}(\tilde{s}_{t+1} | \tilde{a}_t, \tilde{s}_t) := \pi(a_t | \tilde{a}_{\leq t}, a_{<t}) p(s_{t+1} | s_t, a_t), \ \forall t, \tag{5}$$

and

$$\tilde{r}(\tilde{r}_{t+1} | \tilde{s}_{t+1}, \tilde{a}_t, \tilde{s}_t) := \pi(a_t | \tilde{a}_{\leq t}, a_{<t}) r(r_{t+1} | s_{t+1}, a_t, s_t), \ \forall t. \tag{6}$$

As we assume zero-sum adversarial settings, $r_{t+1} = -\tilde{r}_{t+1}$. Note that the victim's policy conditions on its action-observation history, and not just on its last observation, as it contains a sequence detector. As a result, the $\mathcal{I}$-MDP differs from the SA-MDP (Zhang et al., 2021) in that the state $\tilde{s}_t := \{s_t, \tilde{a}_{<t}, a_{<t}\}$, on which the adversarial policy $\pi_{adv}$ is conditioned, is given by the state of the MDP $\mathcal{E}$, as well as the victim's action-observation history.

**Definition 4.7** (Learning $\mathcal{W}$-illusory attacks)**.** Given a fixed victim policy $\pi_v$, a $\mathcal{W}$-illusory attack $\pi_a$ on $\mathcal{E}$ can be learnt by optimising

$$\max_{\pi_a} \mathbb{E}_{\mathcal{I}(\mathcal{E}, \pi_v)} \left[ \sum_{t=0}^{T-1} r_t - \gamma \lambda \mathcal{D} \left[ m_a \left( a_{t-1}, \pi_v(a_{t-1}) \right), \pi_a(s_t, a_{t-1}, \pi_v(a_{t-1})) \right] \right]. \tag{7}$$

Here, we choose the $L_2$-distance, i.e. $\| m_a \left( a_{t-1}, \pi_v(a_{t-1}) \right) - a_t \|_2^2$ for deterministic environments, where $m_a$ is the adversary's world model. For stochastic environments, a possible choice would be $\mathcal{D}_{KL}$, however, alternatively, one could simply employ the maximum-likelihood objective $\log m_a \left( a_t | a_{t-1}, \pi_v(a_{t-1}) \right)$. Non-Markovian temporal correlations in the adversarial policy can be suppressed by restricting the horizon of the adversarial policy's observation space $\tilde{\mathcal{S}}$, which could be detected by the victim (Shi et al., 2020).

### 4.5 Reality feedback

So far we have assumed that the adversary can corrupt all parts of the victim's observation. However, in practice, the AI agent ~~may be able to~~ could receive a limited amount of unperturbed environment ("reality") feedback through robust channels which it ~~may, in principle,~~ can use to act optimally in the presence of an adversary.

**Definition 4.8** (Reality feedback)**.** We define reality feedback $\zeta$ as a part of the victim's observation in $\mathcal{E}'$ that cannot be corrupted by the adversary, i.e. we assume that $\mathcal{Z} := \mathcal{Z}_0 \times \mathcal{Z}_\zeta$, where the adversary has access to $z^0 \in \mathcal{Z}_0$ but not $z^\zeta \in \mathcal{Z}_\zeta$.

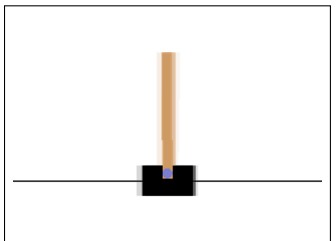 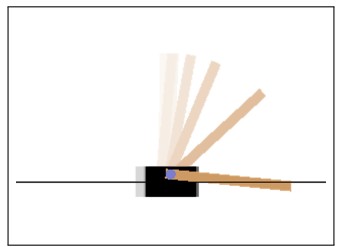 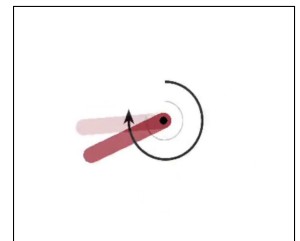

Figure 2: In CartPole, the agent aims to balance the brown pole by adjusting the position of the black cart. In the perfect illusory attack depicted above, the agents observations (left) appear unperturbed while the true system fails (right).

Figure 3: In Pendulum, the agent must apply a torque to stabilize the pendulum upright.

## 5 EXPERIMENTAL EVALUATION

In Section 5.2, we first provide empirical evidence that state-of-the-art attacks on sequential decision-makers can be detected by AI victims with a learnt world model. In Section 5.3 we then provide examples of both perfect illusory attacks (see Definition 4.2), as well as learned $\mathcal{W}$-illusory attacks (see Definition 4.5). In Section 5.4, we give evidence that humans can efficiently detect state-of-the-art adversarial attacks, but ~~may~~ can be unable to detect $\mathcal{W}$-illusory attacks. Lastly, we investigate the effectiveness of state-of-the-art adversarial robustification algorithms against $\mathcal{W}$-illusory attacks (Section 5.5). We find that effective defences against illusory attacks ~~may~~ can require the victim to have access to unperturbed feedback channels (Section 5.5).

### 5.1 EXPERIMENTAL SETUP

We compare illusory attacks against two *adversarial attack benchmarks*, (Kumar et al., 2021, MNP) and (Zhang et al., 2021, SA-MDP) (see Section 3). Together, we consider these attacks to be representative of existing state-of-the-art approaches. Both MNP and SA-MDP attacks require choosing perturbation budgets, which we here fix at $\beta \in \{0.05, 0.2\}$ relative to normalised observation vectors. To allow for fair comparisons, where necessary we also impose these budgets on our generated illusory attacks. We consider two choices of *victim policy*: *naive* policies learnt through train-time return maximisation in the unattacked environment, and *ATLA* policies learnt through co-training (Zhang et al., 2021, ATLA). We evaluate all methods on three simulated *environments*. Gridworld (see Figure 1) involves navigating to a "good" grid cell while avoiding a "bad" grid cell. The OpenAI gym (Brockman et al., 2016) environments CartPole and Pendulum (see Figures 2 and 3) feature continuous control tasks that require balancing a pole with discrete actions, and swinging up a pendulum with continuous torques. In accordance with Section 4.4, we assume that the adversary has access to an exact model of the environment dynamics $m_a$. To ensure *reproducibility*, all results are averaged over five runs with independent random seeds, and we report full results with standard deviations in Table 7.5.1 in the Appendix. We release our code at `anonymized`.

### 5.2 DETECTING STATE-OF-THE-ART ADVERSARIAL ATTACKS WITH LEARNED WORLD MODELS

We demonstrate that existing state-of-the-art adversarial attacks can be detected using learned world models. To this end, we use train-time trajectory rollouts to train a small neural network-based *world model* $\hat{m}_v$ for each environment (see Appendix 7.3). We then evaluate each $\hat{m}_v$ on episodes from a holdout set of test-time trajectories from both the respective attacked and unattacked environment and under both naive and ATLA victim policies (see Section 5.1). To account for the neural network models' finite precision, we regard transitions to be consistent with $m_v$ if they lie within the $5\sigma$ confidence bound, which we computed on a held-out set of train-time transitions.

We find that both MNP and SA-MDP attacks frequently generate world-model inconsistent transitions, which allows for $> 93\%$ of attacked episodes to be detected with high accuracy, given the false-positive rate of $\hat{m}_v$ on unattacked environments of $< 1\%$. Figure 1 gives a graphical illustration of attacks on Gridworld. We provide videos of all attacks in the supplementary material (see Appendix 7.5.3).

| Attack | Budget $\beta$ | Detection accuracy | | Relative adversary score under different defences | | | |
|---|---|---|---|---|---|---|---|
| | | naive | ATLA | none | smoothing | ATLA | ATLA abl. |
| **Pendulum** | | | | | | | |
| SA-MDP (Zhang et al., 2021) | 0.2 | 97% | 93% | 100 % | 87.4 % | 97.9 % | 100 % |
| Illusory (ours) | | 0% | 0% | 86.2 % | 71.7 % | 76.7 % | 75.3 % |
| SA-MDP (Zhang et al., 2021) | 0.05 | 96% | 95% | 100 % | 60.6 % | 95.9 % | 100 % |
| Illusory (ours) | | 1% | 2% | 83.9 % | 58.5 % | 83.5 % | 90.4 % |
| **CartPole** | | | | | | | |
| MNP (Kumar et al., 2021) | | 100% | – | 98.1 % | 97.6 % | | |
| SA-MDP (Zhang et al., 2021) | 0.2 | 99% | 96% | 99.9 % | 93.9 % | 100 % | 99.9 % |
| Illusory (ours) | | 2% | 3% | 100 % | 97 % | 99.9 % | 99.8 % |
| MNP (Kumar et al., 2021) | | 96% | – | 3.1 % | 65.2 % | | |
| SA-MDP (Zhang et al., 2021) | 0.05 | 94% | 100% | 100 % | 76.9 % | 97 % | 98.5 % |
| Illusory (ours) | | 0% | 2% | 100 % | 68.2 % | 97.5 % | 92.6 % |
| **HalfCheetah (MuJoCo)** | | | | | | | |
| SA-MDP (Zhang et al., 2021) | 0.05 | 100% | 100% | 85.4% | tbd % | tbd % | tbd % |
| Illusory (ours) | | 0% | 0% | 32.1% | tbd % | tbd % | tbd % |
| SA-MDP (Zhang et al., 2021) | 0.2 | 100% | 100% | 100 % | tbd % | tbd % | tbd % |
| Illusory (ours) | | 0% | 0% | 48.1% | tbd % | tbd % | tbd % |
| **Hopper (MuJoCo)** | | | | | | | |
| SA-MDP (Zhang et al., 2021) | 0.05 | 100% | 100% | 100% | tbd % | tbd % | tbd % |
| Illusory (ours) | | 0% | 0% | 48.8% | tbd % | tbd % | tbd % |
| SA-MDP (Zhang et al., 2021) | 0.2 | 100% | 100% | 100% | tbd % | tbd % | tbd % |
| Illusory (ours) | | 0% | 0% | 53.4% | tbd % | tbd % | tbd % |

Table 1: Empirical detection accuracy and victim scores for different adversarial attacks and possible defences on Pendulum (top two rows), and CartPole (bottom three rows). Illusory attacks are able to achieve almost perfect undetectability while only sacrificing a small amount of adversarial performance.

## 5.3 LEARNING UNDETECTABLE ADVERSARIAL ATTACKS

We first show concrete examples of non-trivial perfect illusory attacks as defined in Section 4.2. We then empirically investigate $\mathcal{W}$-illusory attacks according to Definition 4.5.

**Perfect illusory attacks.** We find that *Pendulum* and *CartPole* admit perfect, i.e. entirely undetectable, illusory attacks (see Definition 4.2) that significantly impact the victim's performance independently of its policy. For *Cartpole*, such adversarial policies can be constructed by starting off with a coordinate transform $z_0 = -s_0$, and then using the adversary's world model to generate all subsequent observations (see Figure 2). Constructing the adversary's policy proceeds equivalently for Pendulum, as both have symmetric initial state distributions.

~~The existence of strong attacks for two popular benchmark environments may astonish, as effective perfect illusory attacks do not always exist (see Appendix 7.1). This may suggest that robotic control environments of practical interest could be particularly susceptible to undetectable adversarial attacks.~~

**$\mathcal{W}$-illusory attacks.** We now demonstrate how illusory attacks can be learnt rather than analytically constructed. We employ Algorithm 1 (see Appendix 7.5), which implements $\mathcal{W}$-illusory adversarial attacks. $\mathcal{W}$-illusory adversarial attacks trade off between the objectives of detectability and adversarial performance. In all of our environments, we choose our detectability objective to be proportional to the $L_\infty$-norm of the distance between the correct next observation, i.e. the next observation expected by the victim, and the adversarial observation generated by the adversary (cf. Definition 4.7). We find the weighting parameter $\lambda = 10$ using a grid search (ablation in Appendix 7.5.2).

Note that while Algorithm 1 could, in principle, discover the exact perfect illusory attacks presented above for both *Cartpole* and *Pendulum*, we do not find this to be the case in practice. We suggest that this is due the convergence to sub-optimal local equilibria.

| | Environment | | |
|---|---|---|---|
| | both | Pendulum | CartPole |
| $P(false \mid$ no attack$)$ | $34.2 \pm 11.4$ | $31.5 \pm 10.5$ | $37.0 \pm 12.3$ |
| $P(false \mid \mathcal{W} -$ illusory (Ours)$)$ | $\mathbf{32.4 \pm 10.8}$ | $\mathbf{37.0 \pm 12.3}$ | $\mathbf{27.7 \pm 9.3}$ |
| $P(false \mid$ SA-MDP$)$ | $81.4 \pm 27.2$ | $96.3 \pm 32.1$ | $66.7 \pm 22.2$ |
| $P(false \mid$ MNP$)$ | $83.3 \pm 27.8$ | — | $83.3 \pm 27.8$ |

Table 2: Results for the study with human participants. The answer "false" signifies that the participants did not believe that a given video displayed the ground truth environment. Note that participants are not more significantly skilled at detecting illusory attacks over non-attacks.

**Detection.** We now compare and contrast $\mathcal{W}$-illusory attacks, MNP attacks and SA-MDP attacks for perturbation budgets $\beta = 0.05, 0.2$. We observe that illusory attacks are much less likely to be detected by the learned world model $\hat{m}_v$ (see section 5.2) than MNP and SA-MDP attacks (see Table 1). This finding is independent of the perturbation budget $\beta$.

ATLA victims are based on a recurrent neural network policy, enabling them, in principle, to learn an *implicit* model of the environment dynamics during train-time. Co-training with such world model-aware victims, could, in theory, result in world-model consistent adversarial attacks. However, we empirically find this not to be the case: As shown in Table 1, attacks on ATLA agents are detected just as likely as those on naive agents. This underlines the need for explicit world model consistency constraints during adversarial training (see Definition 4.7). We provide videos of all types of attacks for different seeds in the supplementary material (see Appendix 7.5.3).

**Victim performance under different attacks.** We find that the average reward achieved by the victim (Table 1, 4th column) is generally lower for $\beta = 0.2$, and also generally lower for SA-MDP attacks than for $\mathcal{W}$-illusory attacks. This is to be expected, as $\mathcal{W}$-illusory attacks trade off between minimizing the reward of the victim and generating consistent observation sequences, while an SA-MDP attack solely minimizes the victim reward. Given that illusory attacks are significantly more constrained than SA-MDP attacks, we consider it surprising that undetectability can be traded in for only minor decreases in adversarial performance.

## 5.4 FOOLING HUMANS WITH ILLUSORY ATTACKS

We perform a controlled experiment with $n = 10$ human participants in order ~~to collect evidence for our claim that while humans can efficiently detect state-of-the-art adversarial attacks, they cannot detect $\mathcal{W}$-illusory attacks~~ to investigate whether humans unfamiliar with adversarial attacks are able to detect $\mathcal{W}$-illusory attacks. Participants were first shown example videos of both *CartPole* and *Pendulum* trajectories, exposing them to environment-specific dynamics. Participants were then asked to classify a random mixture of videos of test-time trajectories from both unattacked and attacked *CartPole* and *Pendulum* environments; further details in Appendix 7.6.

We found that participants were able to visually classify MNP and SA-MDP attacks from videos with high accuracy, similarly for unattacked trajectories. However, humans struggled to correctly classify $\mathcal{W}$-illusory attacks more accurately than non-attacks (see Table 2) as attacked. We establish the statistical significance of these results using a $z$-test statistic (see Section 7.6), which suggests that humans ~~may~~ can be unable to detect such illusory attacks.

## 5.5 ROBUSTIFYING AGAINST ILLUSORY ATTACKS

Having established the effectiveness of illusory attacks in Sections 5.3 and 5.4, we now investigate how victims casn be robustified. We first consider two standard defences against adversarial attacks, *randomized smoothing* (Kumar et al., 2021) and *adversarial pretraining* (ATLA, (Zhang et al., 2021)). We also add an ablation of ATLA that co-trains the victim with a $\mathcal{W}$-illusory adversaries.

---

[1]For a given budget $\beta$, we define the relative adversary score as the percentage deduction in the expected return of the unattacked victim, normalized by the deduction achieved by the best-performing adversary. As indicated by blank spots, MNP attacks cannot be applied to Pendulum (continuous action space), nor ATLA based agents (LSTM policy).

As summarised in Table 1, we find that defences are generally more effective at restoring victim performance at $\beta = 0.05$ . For $\beta = 0.2$, all defences result in only minor improvements. With only minor variations, these trends hold across all three robustification methods we deployed. Note that the victim performance achieved by applying randomized smoothing for $\beta = 0.2$ in CartPole is equivalent to that of a random policy. Overall, our findings suggest that none of the robustification methods studied are particularly effective against illusory attacks.

**Exploiting reality feedback.** We conclude our empirical investigations by exploring the importance of reality feedback in victim robustification (see Section 4.5). We establish two reality feedback scenarios for CartPole: one where the cart observation is unperturbed and one where the observation of the pole is unperturbed. We find that robustifying the victim agent through pre-training ~~may be essential to enable~~ enables victim agents to learn how to effectively use reality feedback. Our results further suggest that having access to *informative* reality feedback channels can allow for significant robustification. See Appendix 7.7 for further details.

## 6 CONCLUSION AND FUTURE WORK

In this paper, we introduce *perfect illusory attacks*, a novel class of adversarial attacks on sequential decision-makers that is undetectable. Importantly, we show that $\mathcal{W}$-illusory attacks, unlike state-of-the-art adversarial attacks, can fool both AI victim agents with access to a model of the environment dynamics, as well as human supervisors. Our results imply that robustification against illusory attacks is mostly ineffective unless the victim has access to reality feedback, making the provisioning of such hardened feedback channels imperative to designers of safe real-world autonomous or human-in-the-loop decision-making systems.

Future work may investigate illusory attacks in large-scale and partially observable victim environments. Similarly, evaluations on real-world settings would be beneficial, e.g. using sim-to-real methodology. Our work further does not investigate the effect of constraining the number of test-time interactions between adversary and victim, or the effect of world model imperfections. Future work must also develop novel approaches and algorithms for victim robustification against illusory attacks. In particular, such methods should be able to maximise the utility of existing reality feedback channels through test-time information gathering behaviour. Lastly, we believe that a conceptual framework unifying both budgeted and illusory attacks could be of value to the community.

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

## 7 APPENDIX

### 7.1 PROOF ON THE EXISTENCE OF PERFECT ILLUSORY ATTACKS

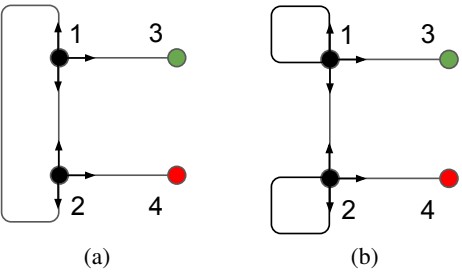

Figure 4: An environment for which perfect illusory attacks do exist (left), and one for which they do not exist (right).

*Proof that perfect illusory attacks exist.* We consider an example MDP (see Figure 4a) where a victim starts in node 1 or 2 each with probability $\frac{1}{2}$ and can go *up*, *down*, or *right* in both states 1 and 2. The episode terminates immediately with a return of 0 should the victim reach state 4. Otherwise, the agent receives a reward of $+1$ if it reaches state 3 within a maximum of 2 steps. The optimal victim policy is therefore to take paths $1 \rightarrow 3$ if starting in state 1, and take one of the two possible paths $2 \rightarrow 1 \rightarrow 3$ otherwise. The victim observes the labelled state graph, as well as its current state label. Clearly, choosing $\pi_{adv}(1) = 2$ and $\pi_{adv}(2) = 1$ constitutes a perfect illusory attack in this environment. □

*Proof that perfect illusory attacks don't always exist.* To show that some environments do not admit perfect illusory attacks, consider the modified environment in Figure 4b. Here, clearly a timestep-conditioned victim policy that takes the action sequence $\langle up, right \rangle$ independently of observations cannot be perfectly attacked. □

### 7.2 FINITE-SAMPLE ILLUSORY ATTACKS

**Definition 7.1** (Finite-Sample Illusory Attacks). A *finite-sample illusory attack* is an adversarial attack that is not detectable by the victim within a given test-time interaction budget $n$. Formally, an $n$-sample illusory attack on an environment $\mathcal{E}$ with victim policy $\pi$ is an adversarial policy $\pi_{adv}$ for which $\nexists D, \nexists \delta > 0$ such that simultaneously

$$D \left( \mathbb{P}(\hat{\tau}_n | \mathcal{E}) || \hat{\mathbb{P}}(\hat{\tau}_n) \right) < \delta, \forall \hat{\tau}_n \sim \mathbb{P} \langle \mathcal{E}, \pi_v \rangle \text{ and } D \left( \mathbb{P}(\hat{\tau}_n | \mathcal{E}) || \hat{\mathbb{P}}(\hat{\tau}_n) \right) \geq \delta, \forall \hat{\tau}_n \sim \mathbb{P} \langle \mathcal{E}', \pi_v, \pi_a dv \rangle . \tag{8}$$

Note that this definition is symmetric between adversary and victim. In practice, the victim will choose a $D, \delta$ such that its average test-time performance is maximised. As the victim is, however, not able to exactly infer $\mathbb{E}_{\mathcal{E}', \pi_v} [R]$ as it does not have access to the adversary's policy, the adversary in turn cannot exactly deduce how exactly the victim chooses $D, \delta$. Therefore, in practice, the adversary might wish to choose an isotropic $D$, such as $D_{KL}$, and construct a policy $\pi_a dv$ that minimises this metric as far as possible given a desired performance-detectability trade-off.

### 7.3 WORLDMODEL TO DETECT ADVERSARIAL ATTACKS

**Setup.** We assume that the victim is trained in the unperturbed MDP $\mathcal{M}_{unperturbed}$ for $k$ episodes of length $n$. During training, the agent records the observed environment transition tuples $t_i = (s_i, a_i, s_{i+1})$ and stores these in a set $\mathcal{D}_{unp} = \{t_i\}_{i=0}^{i=k*(n-1)}$. At test time, the victim executes the learned policy and is attacked by an adversary agent, which replaces the true observation $z_t$ by the adversarial observation $\hat{z}_t$.

**Learning a dynamics model.** We use the the set of unperturbed transitions $\mathcal{D}_{unp}$ to learn an approximate dynamics model $\hat{T}(s_t|s_{t-1}, a_{t-1})$. We implement $\hat{T}$ as a Multi-Layer-Perceptron with two hidden layers of size 10, and train the model with an $l2$ loss for 10 epochs, using a learning rate of 0.001 and the ADAM (Kingma & Ba, 2014) optimizer. We then evaluate the model on a held-out set of the training observations and compute the $5\sigma$ confidence bound on the $l_0$ distance, which we refer to as $c$. We use then use $c$ as a threshold to detect adversarial attacks. The high confidence ensures that the false positive rate remains low.

Note that an adversary with access to $\hat{m}_v$ could attempt to learn universal perturbations (Moosavi-Dezfooli et al., 2017) that simultaneously attack both $\pi_v$ and $\hat{m}_v$. However, adversaries without access to $\hat{m}_v$ may be unable to exploit its imperfections and instead need to assume it is perfect and hence justifying our comparison to solely policy-based MNP/SA-MDP attacks.

**Using a dynamics model to detect adversarial attacks.** We then evaluate the accuracy in detecting adversarial attacks for all scenarios listed in Table 1. More specifically, at every time step, the victim uses the learned world model $\hat{m}_v$ to predict the next observation given the previous observation and action taken. If the $l_0$ distance between the predicted observation and the actual (adversarial) observation is larger than the threshold $c$, the victim classifies the episode as adversarially attacked. Results for this detection mechanism are given in Table 1.

In the six-cell gridworld environment depicted in Figure 1, we similarly evaluated detection accuracy, but only considered naive victim agents. We found that 100% of adversarial attacks could be detected in this environment, while 0% of illusory attacks were detected. We show the resulting sequences of observations and true states in Figure 1.

## 7.4 PERFECT ILLUSORY ATTACKS

We implemented perfect illusory attacks in Pendulum and CartPole by setting the first observation $z_0$ to the negative of the true first state, i.e. $z_0 = -s_0$. We compute all subsequent observations $z_1, ..., z_T$ with the worldmodel $m_a$. As the distribution over initial states is symmetric in both environments, this approach of generating illusory attacks satisfies the conditions of a perfect illusory attack (see Definition 4.2). We provide videos of the generated perfect illusory attacks in the supplementary material in the respective folder. We found that the average reward achieved by the victim under a perfect illusory attack in CartPole was $41.2 \pm 6.8$, while it was $-628.32 \pm 60.1$ in Pendulum. Figure 2 illustrates a perfect illusory attack in CartPole.

## 7.5 LEARNING W-ILLUSORY ATTACKS WITH REINFORCEMENT LEARNING

We next describe the algorithm used to learn $\mathcal{W}$-illusory adversarial attacks and the training procedures used to compute the results in Table 1.

**Setup.** We use the CartPole and Pendulum gym (Brockman et al., 2016) environments. The world model is implemented using the given physics computed for a new environment step. We normalise observations in both environments by the maximum absolute observation. We train the victim with PPO (Schulman et al., 2017) and use the implementation of PPO given in Raffin et al. (2021), while not making any changes to the given hyperparameters. In both environments we train the *naive* victim for 1 million environment steps. We implement the ATLA (Zhang et al., 2021) victim by co-training it with an adversary agent, and follow the original implementation of the authors [2]. We implement the ablation of ATLA (Zhang et al., 2021) that trains the victim with an illusory adversary by merely replacing the ATLA adversary with an illusory adversary, which was implemented as stated in algorithm 7.5. For co-training, we alternate between training the victim and the adversary agent every 400 environment steps. This parameter has likewise been found in a small evaluation study, and was chosen as it yields non-oscillating behaviour. We further investigated different ratios between training steps of the adversary and training steps of the victim, but found that a ratio of one, i.e. equal training of both, yields the most stable and results in co-training.

We implement the illusory adversary agent with SAC (Haarnoja et al., 2018), where we likewise use the implementation given in Raffin et al. (2021). We initially ran a small study and investigated four

---

[2]https://github.com/huanzhang12/ATLA_robust_RL

different algorithms as possible implementations for the adversary agent, where we found that SAC yields best performance and training stability.

We train all adversarial attacks for three million environment steps. We implemented randomized smoothing as a standard defence against adversarial attacks on RL agents, as introduced in Kumar et al. (2021). We use the author's original implementation [3]. In contrast to the authors, we compute the budget with respect to the normalised observations, but otherwise follow the per-step budget definition in Kumar et al. (2021).

---

**Algorithm 1** $\mathcal{W}$-illusory adversarial training

---

**Input:** environment $env$, adversary worldmodel $m_a$, illusory reward weighing parameter $\lambda$, victim policy $\pi_v$, iterations $N$

Initialize $\pi_{adv}^{\psi}$, parametrised be a neural network with parameters $\psi$.

**while** iteration $< N$ **do**

    $s_0 = env()$

    $z_0 = \pi_{adv}^{\psi}(s_0)$                       $\triangleright$ Initial perturbation only depends on initial system state

    $a_0 = \pi_v(clip(z_0))$            $\triangleright$ Adversarial observation is clipped to permissible range

    $s_1, r_1^v, done = env(a_0)$

    $r_1^{adv} = -r_1^v$                            $\triangleright$ No illusory reward at first step

    **while** done=False **do**

        $z_t = \pi_{adv}^{\psi}(s_t, z_{t-1}, a_{t-1})$         $\triangleright$ Adversary conditions on past information

        $a_t = \pi_v(clip(z_t))$

        $s_{t+1}, r_{t+1}^v, done = env(a_t)$

        $r_{t+1}^{adv} = -r_{t+1}^v - \lambda \cdot \|z_t; m_a(z_{t-1}, a_{t-1})\|_{\infty}$       $\triangleright$ Illusory reward added

    **end while**

    $update\ \pi_{adv}^{\psi}$                    $\triangleright$ Adversary policy updated using past experience

**end while**

---

### 7.5.1 FULL RESULTS TABLE FOR CONTRASTING OF DIFFERENT ATTACKS

Table 3: Full results table – top row Pendulum, bottom row CartPole

| attack | budget $\beta$ | Attacks detected | | Victim reward under different defences | | | |
| --- | --- | --- | --- | --- | --- | --- | --- |
| | | naive | ATLA[3] | none | smoothing | ATLA | ATLA abl. |
| SA-MDP (Zhang et al., 2021) | 0.2 | 97% | 93% | $-1387.0 \pm 119.0$ | $-1188.3 \pm 70.4$ | $-1354.6 \pm 107.1$ | $-1428.3 \pm 91.5$ |
| Illusory (ours) | | 0% | 0% | $-1170.1 \pm 67.5$ | $-940.2 \pm 91.6$ | $-1020.4 \pm 50.0$ | $-1029.4 \pm 106.7$ |
| SA-MDP (Zhang et al., 2021) | 0.05 | 96% | 95% | $-797.2 \pm 69.9$ | $-408.4 \pm 146.6$ | $-757.2 \pm 109.3$ | $-722.2 \pm 30.8$ |
| Illusory (ours) | | 1% | 2% | $-638.8 \pm 204.6$ | $-387.8 \pm 115.8$ | $-634.4 \pm 340.7$ | $-634.9 \pm 103.9$ |
| none | | 0% | 0% | $-189.4 \pm 13.4$ | – | $-228.2 \pm 43.8$ | $-220.9 \pm 58.8$ |
| MNP (Kumar et al., 2021) | | 100% | – | $18.3 \pm 20.8$ | $20.8 \pm 8.7$ | – | – |
| SA-MDP (Zhang et al., 2021) | 0.2 | 99% | 96% | $9.3 \pm 0.1$ | $39.01 \pm 10.7$ | $9.2 \pm 0.1$ | $9.7 \pm 0.6$ |
| Illusory (ours) | | 2% | 3% | $9.0 \pm 0.3$ | $23.9 \pm 3.3$ | $9.6 \pm 0.6$ | $10.02 \pm 1.2$ |
| MNP (Kumar et al., 2021) | | 96% | – | $485.0 \pm 33.5$ | $180.3 \pm 33.6$ | – | – |
| SA-MDP (Zhang et al., 2021) | 0.05 | 94% | 100% | $9.4 \pm 0.2$ | $122.5 \pm 54.3$ | $24.2 \pm 7.3$ | $16.8 \pm 8.3$ |
| Illusory (ours) | | 0% | 2% | $9.3 \pm 0.1$ | $165.4 \pm 46.3$ | $21.4 \pm 6.0$ | $45.4 \pm 56.5$ |
| none | | 0% | 0% | $500.0 \pm 0$ | – | $500.0 \pm 0$ | $500.0 \pm 0$ |

Table 7.5.1 shows all results for all scenarios evaluated; we report mean and standard deviation. Note that each scenario was evaluated for five random seeds; Table 1 in the main paper is derived from Table 7.5.1.

### 7.5.2 ABLATION: INFLUENCE OF LAMBDA ON RESULTS

We investigated the influence of the $\lambda$ parameter in a small study. Generally, the $\lambda$ parameter determines the trade off of the illusory adversary between generating consistent observation sequences, and minimizing the victim's reward. We considered $\lambda \in \{1, 10, 100\}$, with $\beta = 0.2$. Table 4 shows that smaller weighing parameters $\lambda$ generally lead to lower victim rewards, which is in line with the fact that for $\lambda = 0$, the objective of the adversary is solely to minimise the victim reward.

---

[3] https://openreview.net/forum?id=mwdfai8NBrJ

Table 4: Reward achieved by the victim across different weighing parameter $\lambda$.

| | Weighing parameter $\lambda$ | | |
|---|---|---|---|
| Environment | 1 | 10 | 100 |
| CartPole | $9.0 \pm 0.3$ | $9.0 \pm 0.3$ | $22.2 \pm 3.7$ |
| Pendulum | $-1382.87 \pm 158.3$ | $-1170.1 \pm 67.5$ | $-616.1 \pm 120.0$ |

Table 5: Results from our study with human participants.

| | Environment | | |
|---|---|---|---|
| | both | Pendulum | CartPole |
| $P(false \mid$ no attack$)$ | $\mathbf{34.2} \pm 11.4$ | $31.5 \pm 10.5$ | $37.0 \pm 12.3$ |
| $P(false \mid$ **SA-MDP**$)$ | $81.4 \pm 27.2$ | $96.3 \pm 32.1$ | $66.7 \pm 22.2$ |
| $P(false \mid \mathcal{W} -$ illusory$)$ | $\mathbf{32.4} \pm 10.8$ | $37.0 \pm 12.3$ | $27.7 \pm 9.3$ |
| $P(false \mid$ MNP$)$ | $83.3 \pm 27.8$ | | $83.3 \pm 27.8$ |
| $P(false)$ | $54.2 \pm 18.1$ | $54.9 \pm 18.3$ | $53.7 \pm 17.9$ |

### 7.5.3 VIDEOS OF ALL ADVERSARIAL ATTACKS

We provide videos for different seeds for all adversarial attacks in the supplementary material. The folders are named respectively. Note that the left side shows the environment as observed by the victim agent, while the right side shows the true environment state. Please further note that the episode in CartPole terminates once the angle has crossed a specific threshold, which is why the episodes might appear to be shorter than otherwise expected. All videos were generated for a budget $\beta = 0.2$.

### 7.6 HUMAN STUDY

We performed a controlled experiment with $n = 10$ human participants. All participants were graduate-level university students, while none had prior knowledge about the objective of the study. Participants participated voluntarily; we estimate the time needed per participant was around 15 minutes. Participants were handed a slide show which contained all relevant information. This slide show is included in the supplementary material in the respective folder. We further add the sheet with ground truth labels for all video sequences.

After consenting to participate, participants were provided with the slide show and an online sheet to indicate their answers. The study was self-paced and done by the participants without further supervision. We further advised participants not to do the study if they would not feel comfortable at any point in time.

The first 18 videos in the slide show are 100-frame long sequences of the Pendulum environment, i.e. six each for either no adversarial attack, a $\mathcal{W}$-illusory attack or an SA-MDP attack (Zhang et al., 2021). The subsequent 24 10-frame long sequences of the CartPole environment include the same attacks as for Pendulum, plus an additional six sequences under an MNP attack (Kumar et al., 2021). Note that MNP attacks cannot be applied to the Pendulum environment, as it has a continuous action space. For all attacks, the budget $\beta$ was 0.2. Within both groups, the order of the videos shown was randomised. Subjects had to choose for each video between $correct$ and $false$, where $false$ indicates that subjects find that the system observed in the respective video is not the same as the one observed in the introductory video. Table 5 includes all empirically estimated probabilities of classifying a segment from a given class as $false$, split up also by the different environments CartPole and Pendulum.

**Result analysis.** We find that subjects reported an unexpectedly high proportion (one third) of unattacked sequences as $false$. Subjects classified sequences which were under an MNP or SA-MDP (Zhang et al., 2020) attack as not from the same system with a probability higher than 80%, while categorizing illusory attacks as $false$ with a probability of only 32.4%. We computed the double-sided z-statistics and were able to reject both the hypothesis that $p(false \mid$ SA-MDP$) = p(false \mid$ none$)$ and the hypothesis that $p(false \mid$ MNP$) = p(false \mid$ none$)$ for $\alpha = 0.05$, while the hypothesis that $p(false \mid \mathcal{W}$-illusory$) = p(false \mid$ none$)$ cannot be rejected. We conclude that subjects were able to

Table 6: Reward achieved by victim for different reality feedback scenarios.

| reality feedback | Victim agent | |
|---|---|---|
| | naive | ATLA abl. |
| pole | $9.84 \pm 0.1$ | $\mathbf{182.44} \pm 36.9$ |
| cart | $8.83 \pm 0.3$ | $15.54 \pm 6.6$ |

distinguish SA-MDP and MNP attacks from unattacked sequences, while being unable to distinguish illusory attacks from unattacked sequences.

## 7.7 REALITY FEEDBACK

**Setup.** We evaluate the importance of realism feedback in the *CartPole* environment by investigating two possible scenarios. Note that the observation in *CartPole* is given as a four-dimensional vector of the pole angle and angular velocity, as well as cart position and velocity. In the first test scenario, the victim correctly observes the pole, while the adversary can perturb the observation of the cart; the second scenario is vice versa. We investigate two test cases for each scenario: First, attacking a naive victim, and second, attacking an agent pretrained with co-training.

**Results and discussion.** Table 6 shows that the reward achieved by the victim is generally higher when pretrained with co-training. We hypothesize that this pretraining enables the agent to learn how to utilize the reality feedback effectively. The achieved victim performance when reality feedback contains information about the *pole* is more than 10 times larger than when containing information on the *cart* instead. This seems intuitive, as the observation of the pole appears much more useful for the task of stabilizing the pole, and underlines the importance of equipping agents with strong reality feedback channels.

