# OpenReview forum: "Illusory Adversarial Attacks on Sequential Decision-Makers and Countermeasures"
_ICLR.cc/2023/Conference — Submitted to ICLR 2023_

### Official Review · Reviewer_G6kZ · 2022-10-23

**Confidence:** 4
**Correctness:** 2
**Technical Novelty And Significance:** 3
**Empirical Novelty And Significance:** 2
**Recommendation:** 5

**Clarity, Quality, Novelty And Reproducibility:**

This paper is clearly written. The problem setup is relatively novel. The code and hyperparameter settings are provided.

**Strength And Weaknesses:**

Strengths:

1. The idea of statistically consistant attacks is interesting, and to my knowledge, novel in the literature.
2. The paper provides theoretical justifications that perfect illusory attacks exist for some but not all policy-environment pairs.
3. Provided human study are useful for understanding the detectability of adversarial attacks.
4. The proposed algorithm makes intuitive sense, and the empirical results on Cartpole and Pendulum do show the effectiveness of the algorithm.

Weaknesses:

1. I do not agree with some claims made by the paper. In particular, most existing adversarial attacks are imperceptible to humans [1,2], which is an important motivation of adversarial attacks. Most literature of adversarial RL studies the adversarial perturbations on observation, which lies in a high-dimensional space (e.g. images [2,3,4]). These perturbations are usually undetectable by humans, unless special care is taken.
2. The assumption of an exact world model is not very realistic, especially in the senarions where adversarial attacks are concerning. In simulators, we may have an exact world model. However, it is less possible and also less dangerous that a local simulator is attacked. Adversarial examples are more critical during the interaction with real-world environments where observation may be noisy and exposed to outside attackers. Under these real-world scenarios, the access to a world model is unrealistic. Even learning a good model can be challenging in these environments.
3. The experiments are on simple environments. Can the authors also provide results on larger scale environments like Atari games, or at least MuJoCo environments?
4. Some related works are missing. For example, [4] proposes a stronger adversarial attack that SA-MDP [3]. Can the authors compare the proposed attack with [4]?



[1] Goodfellow, Ian J., Jonathon Shlens, and Christian Szegedy. "Explaining and harnessing adversarial examples."

[2] Huang, Sandy, Nicolas Papernot, Ian Goodfellow, Yan Duan, and Pieter Abbeel. "Adversarial attacks on neural network policies."

[3] Zhang, Huan, Hongge Chen, Chaowei Xiao, Bo Li, Mingyan Liu, Duane Boning, and Cho-Jui Hsieh. "Robust deep reinforcement learning against adversarial perturbations on state observations.

[4] Sun, Yanchao, Ruijie Zheng, Yongyuan Liang, and Furong Huang. "Who is the strongest enemy? towards optimal and efficient evasion attacks in deep rl."

**Summary Of The Paper:**

This paper studies adversarial attacks on sequential decision-making policies, with a focus on *statistically undetectable* attacks. The authors assume exact knowledge of the world model, and introduce a novel class of adversarial attacks called illusory attacks, which are consistant with the world dynamics and thus more stealthy. This paper formulates the illusory attacks, and propose a feasible learning algorithm, W-illusory attacks, to generate illusory attacks. Experiments on simple control tasks show that the proposed attack and less detectable to humans and AI agents than state-of-the-art attacks.

**Summary Of The Review:**

The formulation and methods proposed by the paper are interesting. But I do have some concerns on the significance of the problem, as well as the realisticity of the key assumptions. In short, it is not clear to me whether the illusory attack and the world model assumption is practical in real-world scenarios and high-dimensional environments.

---

> ### Author Response · Authors · 2022-11-10
> **Response**
>
> >Summary Of The Paper:
> This paper studies adversarial attacks on sequential decision-making policies, with a focus on statistically undetectable attacks. The authors assume exact knowledge of the world model, and introduce a novel class of adversarial attacks called illusory attacks, which are consistant with the world dynamics and thus more stealthy. This paper formulates the illusory attacks, and propose a feasible learning algorithm, W-illusory attacks, to generate illusory attacks. Experiments on simple control tasks show that the proposed attack and less detectable to humans and AI agents than state-of-the-art attacks.
>
> >Strength And Weaknesses:
> >Strengths:
>
> >The idea of statistically consistant attacks is interesting, and to my knowledge, novel in the literature.
> >The paper provides theoretical justifications that perfect illusory attacks exist for some but not all policy-environment pairs.
> >Provided human study are useful for understanding the detectability of adversarial attacks.
> >The proposed algorithm makes intuitive sense, and the empirical results on Cartpole and Pendulum do show the effectiveness of the algorithm.
> Weaknesses:
> >I do not agree with some claims made by the paper. In particular, most existing adversarial attacks are imperceptible to humans [1,2], which is an important motivation of adversarial attacks.
>
> A4.1: Prior attacks are imperceptible to humans due to the budget constraint, e.g. by considering low-amplitude noise in an image. This previous work [1,2] focuses on observations at a single time step, such as a single image. In contrast, our work considers adversarial attacks on environment interactions that generate *sequences* of observations. This setting requires different considerations regarding perceptibility (by humans or learning systems) from single-image attacks. Specifically, we do not ask whether the attacked observation can be distinguished from the original one, but whether the distribution of trajectories matches between the attacked and original environment.
>
> >Most literature of adversarial RL studies the adversarial perturbations on observation, which lies in a high-dimensional space (e.g. images [2,3,4]). These perturbations are usually undetectable by humans, unless special care is taken.
> >The assumption of an exact world model is not very realistic, especially in the senarions where adversarial attacks are concerning. In simulators, we may have an exact world model. However, it is less possible and also less dangerous that a local simulator is attacked. Adversarial examples are more critical during the interaction with real-world environments where observation may be noisy and exposed to outside attackers. Under these real-world scenarios, the access to a world model is unrealistic. Even learning a good model can be challenging in these environments.
>
> A4.2: Thank you for this remark. As we stated in section 4.2, it is not required that the victim has access to a perfect world model. For example, the victim may only have access to an abstract model of the environment dynamics that, by itself, would not be sufficient for planning, but may nevertheless allow the victim to perform effective checks on trajectory consistency. In general, the required accuracy of the attacker’s world model depends on the accuracy of the victim’s world model.
>
> >The experiments are on simple environments. Can the authors also provide results on larger scale environments like Atari games, or at least MuJoCo environments?
>
> A4.3: As suggested by the reviewers, we will add additional experiments on MuJoCo by the end of the weekend. Initial results can be retrieved fully anonymously here:
> https://drive.google.com/file/d/1ffNtT3RtvEQRAX3D9A2mezdappCVq-Ku/view?usp=sharing
>
> >Some related works are missing. For example, [4] proposes a stronger adversarial attack that SA-MDP [3]. Can the authors compare the proposed attack with [4]?
>
> A4.4: Thank you for this remark, we added [4] to the related work. [4] investigates adversarial attacks on high-dimensional observation spaces, by improving the scalability of the method proposed in [3]. In contrast to our work, [3] and [4] do not consider statistical indistinguishability.

---

> > ### Comment · Reviewer_G6kZ · 2022-12-07
> > **Thank you for the response**
> >
> > I think the problem and the idea proposed by this paper are both interesting. It helps the community to consider multiple attack models other than a common $l_p$ model.
> >
> > However, the current paper still has several main limitations that make the significance of the study not very satisfactory. First, I still have concerns on the world model. Even if the authors claim that the world model does not need to be exact, there is no theoretical proofs, nor empirical evidence. How "rough" can the world model be? Second, the environments are too simple compared to existing works. The potential reason could be the hardness of building a good world model for large-scale environments. It raises concerns on whether the proposed method is practical in real world problems.
> >
> > In the provided google drive, I can only see the "no attack" videos, while the attacked ones are missing. So I still do not know whether the method extends to larger environments.
> >
> > Based on the above reasoning, I will maintain my rating. But I would be happy to see a better version of this paper in the future.

---

> > > ### Author Response · Authors · 2022-12-08
> > > **Response**
> > >
> > > Dear reviewer,
> > >
> > > Thank you for your comment. We have inspected the google drive link and found that it should work well.
> > >
> > > __Please wait until the video is at 0:17 (17 seconds) to see SA-MDP attacks. At 0:40 (40 seconds), you can observe illusory attacks.__
> > > __These videos demonstrate the scalability to larger environments.__
> > >
> > > Please let us know if this still does not work for you.
> > >
> > > We will address the other aspects of your response within the next hours.

---

> > > > ### Comment · Reviewer_G6kZ · 2022-12-08
> > > > **Update**
> > > >
> > > > Thank you for the explanation. Sorry I did not realize that the 3 rows are played sequentially in the video. Now I can see the video. It is indeed interesting.
> > > >
> > > > For the world model. I do not agree that analysis is impossible on how "rough" the model should be.
> > > > - In experiment, it is straightforward to investigate the relation between model accuracy and attack success rate. For example, you may consider using different checkpoints during the training of the world model to launch attacks, and draw a plot showing how model error rates is related to the attack performance. It will be more promising if the attack is successful even if the trained world model is less accurate.
> > > > - In theory, there are a lot of model-based RL papers [1,2] analyzing the relation between model estimation error and performance gap. So I think it is feasible to show some theoretical analysis for the proposed attack. It does not need to be complicated. It would be helpful to have some insights on how the performance of the obtained attacker is different from the optimal one when the model $\hat{\mathcal{E}}$ is imperfect and have model error rate $\epsilon_m$.
> > > > - It is true that learned dynamics models can help a lot even in large-scale environments such as video games. But please note that the papers you cited **do not learn the original dynamics $P$ (in the raw observatiton space)**. They instead learn the transitions in the hidden state space or learn the inverse dynamics model, which is just because learning the full dynamics model for high-dim observations is hard. In contrast, the method proposed in this paper requires a world model that is operated on the observation space, which is hard to construct in practice.
> > > >
> > > > In summary, I think this paper studies an interesting problem. I am okay with accepting it.  But I am still hesitating because the practical contribution of the current paper seems limited to me. I would expect more empirical evidence or theoretical insights showing that the proposed attack is resilient to model errors, or can work for larger-scale tasks such as walker2d, ant, or even Atari games.
> > > >
> > > > [1] Janner, Michael, et al. "When to trust your model: Model-based policy optimization." Advances in Neural Information Processing Systems 32 (2019).
> > > >
> > > > [2] Grimm, Christopher, et al. "The value equivalence principle for model-based reinforcement learning." Advances in Neural Information Processing Systems 33 (2020): 5541-5552.

---

> > > > ### Author Response · Authors · 2022-12-09
> > > > **Response to update**
> > > >
> > > > >In experiment, it is straightforward to investigate the relation between model accuracy and attack success rate. For example, you may consider using different checkpoints during the training of the world model to launch attacks, and draw a plot showing how model error rates is related to the attack performance. It will be more promising if the attack is successful even if the trained world model is less accurate.
> > > >
> > > > We assume the reviewer is interested in varying both the accuracy of the victim's and the attacker's world models relatively to each other. We agree that this is an interesting question for practical attacks, however, we also expect this to strongly depend on a) the task and b) assumptions on practical limitations of both victims and attackers.  Nevertheless, we are happy provide the sample empirical evaluation proposed by the reviewer with the camera ready.
> > > >
> > > > >In theory, there are a lot of model-based RL papers [1,2] analyzing the relation between model estimation error and performance gap. So I think it is feasible to show some theoretical analysis for the proposed attack. It does not need to be complicated. It would be helpful to have some insights on how the performance of the obtained attacker is different from the optimal one when the model E is imperfect and have model error rate e.
> > > >
> > > > We agree that a theoretical analysis of the relation between model estimation error and performance gap would be interesting, albeit perhaps slightly out of scope. We are not currently sure how to directly extend the frameworks of [1,2] directly to W-illusory attacks as their learning objectives differ from the ones investigated in [1,2]. We are happy to further look into this for the camera ready version, but cannot promise success at this point.
> > > >
> > > > >It is true that learned dynamics models can help a lot even in large-scale environments such as video games. But please note that the papers you cited do not learn the original dynamics P(in the raw observatiton space). They instead learn the transitions in the hidden state space or learn the inverse dynamics model, which is just because learning the full dynamics model for high-dim observations is hard. In contrast, the method proposed in this paper requires a world model that is operated on the observation space, which is hard to construct in practice.
> > > >
> > > > The reviewer is correct to point out that our current illusory attack framework is explicitly demanding observation-space world models. However, our framework can be easily adjusted to allow for world models of transitions in the hidden state space or inverse dynamics. For example, we can readily map high-dim observations to their hidden states and then perform regularization directly on the hidden states. We will surgically add the required formalisms to our camera ready version.

---

> > > > ### Author Response · Authors · 2022-12-09
> > > > **score**
> > > >
> > > > Dear reviewer, given the rebuttal clarifications and additional experiments, would you be willing to increase your score?

---

> > > ### Author Response · Authors · 2022-12-08
> > > **Response to comments**
> > >
> > > > First, I still have concerns on the world model. Even if the authors claim that the world model does not need to be exact, there is no theoretical proofs, nor empirical evidence. How “rough” can the world model be?
> > >
> > > Expanding on answer A4.2: As stated, the required accuracy of the attacker’s world model depends on the accuracy of the victim’s world model. Hence, it is not possible to state how "rough" the world model may be, as this entirely depends on the world model of the victim agent. Further, we did not find any theoretical analyses that compares the correctness of two worldmodels in literature, and do not see a possibility (nor necessity) to derive a meaningful theoretical comparison.
> > >
> > > > Second, the environments are too simple compared to existing works. The potential reason could be the hardness of building a good world model for large-scale environments. It raises concerns on whether the proposed method is practical in real world problems.
> > >
> > > We would like to refer to "Mastering atari, go, chess and shogi by planning with a learned model." (Schrittwieser, et al., 2020) and "Video PreTraining (VPT): Learning to Act by Watching Unlabeled Online Videos" (Baker, et al. 2022) who demonstrate __successful learned world models in the highly complex environments Minecraft, Atari and Go__. Please also check the drive link again (see comment above) to see results of our work in more complex domains.

---

### Official Review · Reviewer_pRsS · 2022-10-25

**Confidence:** 4
**Correctness:** 3
**Technical Novelty And Significance:** 3
**Empirical Novelty And Significance:** 3
**Recommendation:** 6

**Clarity, Quality, Novelty And Reproducibility:**

Please see my detailed comments above. Below I summarize some of the points related to quality, clarity and originality.

- Quality: I believe that the attack model studied in this work is interesting, but given that the evaluation is primarily based on experiments, more environments could be added to the experiments test-bed. Regarding the reproducibility, the simulation-based results are well documented. On the other hand, it would be good to extend the description of the human-subject experiment, e.g., by adding the recruitment protocol, and indicate whether the study had received an IRB approval.

- Clarity: The paper is overall clearly written. That said, some part of the paper are not entirely clear as I indicated above. If these are typos, unfortunately, there seem to be quite a few of them and they significantly impact the correctness of the results...

- Originality: The attack model studies in the paper seems quite novel. I also like the fact that the paper utilizes human-subject experiment to test the detectability of some of adversarial attacks, which is rather novel, or at least not that common in this line of work.

**Details Of Ethics Concerns:**

As I wrote in my comments above, the paper has a human-subject study, but does not seem to report whether this study had received an IRB approval.

**Strength And Weaknesses:**

**Strengths of the paper**:
- To my knowledge, the attack models considered in this work has not been studied in the literature on test-time attacks against RL agents. The attack models are well motivated and complement those from prior work. Instead of focusing on $L_p$ norm-based attack models, the paper advocates models that are statistically undetectable.
- The paper introduces a formal framework for studying these attack models, as well as an optimization problem for finding an optimal illusory attacks. The optimization problem aims to minimized the victim's return, while minimizing a distance function that measures the inconsistency between generated trajectories and the environment dynamics.
- The paper also conducts a human-subject experiment to test the detectability of this attack model via visual inspection. This validation techniques appears to be novel when it comes to adversarial attacks on RL agents.

---
**Weaknesses of the paper**:
- The experiments are primarily based on two simple environments, Pendulum and CartPole. In contrast, prior work, e.g., Zhang et al. 2021, has studies more complex environments, such as MuJoCo. More experiments would be useful in order to understand the scalability of the approach. Also, given that one the experiments involves human-subjects, IRB may be required; the paper doesn't seem to report if the study has an IRB approval.
- In general, it is not clear what are the computational properties of the proposed optimization framework. The I-MDP model does not scale well with the time horizon, and given that the experiments are only based on two simple environments, it is not clear how practical this approach is. Additionally, the attack optimization problem (7) seems to require the environment/world model. Some discussion on the practicality of the approach would be useful to have.
- Some parts of the formal framework are not entirely clear and may contain typos. Firstly, I don't fully understand why rewards are not included in trajectory \tau when one measure consistency with the true environment, nor why \tilde S does not include rewards. It's not immediately clear to me that the victim cannot detect this attack by inspecting received rewards. Some discussion on this would be useful. Secondly, definition 4.1 uses $\mathcal E '$ but does not specify it. Thirdly, Eq. (3), (5), (6) and (7) may not be precise. E.g., why do we minimize $-\lambda D()$ in (3)? Moreover, transition probabilities $p(.)$ should be functions of $s$, not $\tilde s$, but in (5) we have $p(\tilde s_{t+1}|\tilde s_{t}, a_{t})$. Similarly, I don't understand how we obtain (6) since rewards function $r$ should take elements from $S$ not $\tilde S$; it is also not clear why this equality holds as we have $\tilde r_{t+1}$ on LHS and $r_{t+1}$ on RHS. In (7), why do we have $r_{t}$ and not $\tilde r_{t}$ and why does $\pi_a$ depend only on $a_{t-1}$ and $\pi_v(a_{t-1})$ but not the whole history? The notation also seems to be ambiguous. E.g., is $a_t$ the action of the attacker or the victim, and does $\pi_v(a_{t-1})$ denote stochastic or deterministic policy?

**Summary Of The Paper:**

This paper studies test-time attacks on reinforcement learning agents. It focuses on attacks that are statistically undetectable, and proposes novel attack models that aim to preserve consistency of trajectories with the environment dynamics. The paper develops a new optimization framework for generating such attacks and experimentally validates the effectiveness of these attacks, called illusory attacks. The experiments test the efficacy of the proposed approach in terms of: i) detectability of adversarial attacks via statistical consistency checks, ii) detectability of adversarial attacks via visual inspection (i.e., human-subject studies), and iii) susceptibility of robustly trained RL agents to adversarial attacks. The experimental results indicate that the proposed attack approach yields lower detectability rates compared to prior works.

**Summary Of The Review:**

Overall, I enjoyed reading the paper. The core idea seem quite intriguing and novel. Having said that, I believe that the paper could benefit from having more experiments (i.e., additional environments), and that it could better explain the limitations of this work and the practicality of the proposed approach. Apart from that, some parts of the paper could be polished.

---

> ### Author Response · Authors · 2022-11-10
> **Part 2**
>
>
> >Also, given that one the experiments involves human-subjects, IRB may be required; the paper doesn't seem to report if the study has an IRB approval.
>
> A3.2: Thank you for pointing this out. We will attach the IRB approval to the camera-ready version.
>
> >In general, it is not clear what are the computational properties of the proposed optimization framework. The I-MDP model does not scale well with the time horizon, and given that the experiments are only based on two simple environments, it is not clear how practical this approach is.
>
> A3.3: We are not entirely sure we understand the reviewers question correctly, we assume that the reviewer refers to the scalability of the state-action history of the I-MDP model. We expect that solving I-MDPs scales computationally similarly to solving POMDPs. Note that recurrent policies have been successful at solving large POMDPs [1,2,3].
>
> >Additionally, the attack optimization problem (7) seems to require the environment/world model. Some discussion on the practicality of the approach would be useful to have.
>
> A3.4: We assume that the reviewer is referring to the practicality of the attacker having access to a world model. We here adopt the common assumption [Zhang et al., 2021] that the attacker has access to the environment (in order to implement the attack), which allows it to estimate a world model. In general, the accuracy of the attacker’s world model required for a successful illusory attack depends on the accuracy of the victim’s world model.
>
> >Some parts of the formal framework are not entirely clear and may contain typos. Firstly, I don't fully understand why rewards are not included in trajectory \tau when one measure consistency with the true environment, nor why \tilde S does not include rewards. It's not immediately clear to me that the victim cannot detect this attack by inspecting received rewards. Some discussion on this would be useful.
>
> A3.5: Thank you for this remark, we have updated the notation accordingly. We assume a test-time adversarial attack on the victim agent, hence the victim agent does not observe the reward signal at test-time (Zhang et al., 2021, Kumar et al., 2021). Imagine a robot trained in simulation, which would likewise not observe the reward during deployment in a real-world scenario.
>
> >Secondly, definition 4.1 uses
> E but does not specify it. Thirdly, Eq. (3), (5), (6) and (7) may not be precise. E.g., why do we minimize [...]
>
> A3.6: Thank you for these remarks. We have addressed the issues in the updated version of the paper.
>
> >Clarity, Quality, Novelty And Reproducibility:
> >Please see my detailed comments above. Below I summarize some of the points related to quality, clarity and originality.
>
> >Quality: I believe that the attack model studied in this work is interesting, but given that the evaluation is primarily based on experiments, more environments could be added to the experiments test-bed.
>
> A3.7: Please see answer A3.1 regarding additional experiments.
>
> >Regarding the reproducibility, the simulation-based results are well documented. On the other hand, it would be good to extend the description of the human-subject experiment, e.g., by adding the recruitment protocol, and indicate whether the study had received an IRB approval.
>
> A3.8: Thank you for pointing this out. We will add the IRB approval and recruitment protocol to the camera-ready version.
>
> >Clarity: The paper is overall clearly written. That said, some part of the paper are not entirely clear as I indicated above. If these are typos, unfortunately, there seem to be quite a few of them and they significantly impact the correctness of the results...
> >Originality: The attack model studies in the paper seems quite novel. I also like the fact that the paper utilizes human-subject experiment to test the detectability of some of adversarial attacks, which is rather novel, or at least not that common in this line of work.
>
> [1] Oriol Vinyals, Igor Babuschkin, Wojciech M Czarnecki, Michael Mathieu, Andrew Dudzik, Juny- oung Chung, David H Choi, Richard Powell, Timo Ewalds, Petko Georgiev, et al. Grandmaster level in StarCraft II using multi-agent reinforcement learning. Nature, 575(7782):350–354, 2019.
>
> [2] Noam Brown and Tuomas Sandholm. Superhuman AI for multiplayer poker. Science, 365(6456): 885–890, 2019.
>
> [3] Baker, Bowen, et al. "Video pretraining (vpt): Learning to act by watching unlabeled online videos." *arXiv preprint arXiv:2206.11795* (2022).

---

> > ### Comment · Reviewer_pRsS · 2022-12-07
> > **Thank you for your response**
> >
> > Thank you for your response. The rebuttal has addressed some of my concerns, so I will increase my score accordingly.

---

> ### Author Response · Authors · 2022-11-10
> **Response Part 1**
>
>
> >Summary Of The Paper:
> >This paper studies test-time attacks on reinforcement learning agents. It focuses on attacks that are statistically undetectable, and proposes novel attack models that aim to preserve consistency of trajectories with the environment dynamics. The paper develops a new optimization framework for generating such attacks and experimentally validates the effectiveness of these attacks, called illusory attacks. The experiments test the efficacy of the proposed approach in terms of: i) detectability of adversarial attacks via statistical consistency checks, ii) detectability of adversarial attacks via visual inspection (i.e., human-subject studies), and iii) susceptibility of robustly trained RL agents to adversarial attacks. The experimental results indicate that the proposed attack approach yields lower detectability rates compared to prior works.
>
> >Strength And Weaknesses:
> >Strengths of the paper:
>
> >To my knowledge, the attack models considered in this work has not been studied in the literature on test-time attacks against RL agents. The attack models are well motivated and complement those from prior work. Instead of focusing on LP norm-based attack models, the paper advocates models that are statistically undetectable.
> The paper introduces a formal framework for studying these attack models, as well as an optimization problem for finding an optimal illusory attacks. The optimization problem aims to minimized the victim's return, while minimizing a distance function that measures the inconsistency between generated trajectories and the environment dynamics.
> The paper also conducts a human-subject experiment to test the detectability of this attack model via visual inspection. This validation techniques appears to be novel when it comes to adversarial attacks on RL agents.
> Weaknesses of the paper:
>
> >The experiments are primarily based on two simple environments, Pendulum and CartPole. In contrast, prior work, e.g., Zhang et al. 2021, has studies more complex environments, such as MuJoCo. More experiments would be useful in order to understand the scalability of the approach.
>
> A3.1: Following the reviewer's request, we present additional experiments on two MuJoCo environments (anonymous link to initial results: https://drive.google.com/file/d/1ffNtT3RtvEQRAX3D9A2mezdappCVq-Ku/view?usp=sharing).
> Our experiments show that perfect illusory attacks exist for both of them, and we suggest that the associated SA-MDP attacks are easily detectable.
> We will further complement results, also for W-illusory attacks, throughout the coming weekend.

---

### Official Review · Reviewer_58cU · 2022-11-01

**Confidence:** 4
**Correctness:** 3
**Technical Novelty And Significance:** 2
**Empirical Novelty And Significance:** 2
**Recommendation:** 5

**Clarity, Quality, Novelty And Reproducibility:**

The main novelty of the paper is to propose an attack algorithm for decision-making agents that is consistent with the unperturbed environment dynamics. The authors try to motivate why statistically undetectable attacks are important. I feel that the writing of the paper can be improved to make the work clearer. I have noted my comments in the first section. I think more evaluations on complex environments are required since the paper focuses on an empirical attack algorithm.

**Strength And Weaknesses:**

Strengths:
1. Focuses on developing attacks that are consistent with the dynamics of the environment.
2. Develops a novel class of illusory attacks that are consistent with the dynamics. Introduces the concept of statistical indistinguishability for stochastic control processes.
3. Well-structured sections. The supplementary material contains all the relevant videos related to human study.

Weaknesses:
1. I think that the human study setup is biased. The samples given in the supplementary show that it's easy to detect attacked vs. unattacked videos. All the unattacked videos labeled in the study balance the pendulum or cart pole perfectly, while all the attacked videos do not do that. Hence, it should be easy to detect attacked vs. unattacked videos for these simple environments.
2. I think illusory vs. unattacked videos can also be figured out without much difficulty. For example, only in illusory attacks the cart pole moves very quickly. In the rest of the attacks/ unattacked videos, the cart poles do not move as fast as in illusory attack videos.
3. Given the above weaknesses, I am not sure why the detection accuracies are low for the illusory attacks in Table 1. What is the naive detection algorithm that is used for Table 1? Also, W-illusory attacks are not "statistically indistinguishable", unlike perfect illusory attacks, right?
4. I am not convinced if good W-illusory attacks should always exist. Pendulum and cart pole are very simple environments. I would suggest adding experiments with more complex environments. Not sure about transferability to other tasks.
5. Reality feedback is a practical and obvious way to defend against illusory attacks. Hence, the attack proposed in this paper does not appear to be strong.

Response to rebuttal
================

Thanks for the rebuttal; however, some of my concerns remain. Could you please try to address these?

1. I am not convinced about illusory attacks not being detectable to "attentive human supervisors". I think a non-expert in RL can figure out attacked vs. unattacked in cases of illusory attacks. Quoting from my review -- "For example, only in illusory attacks the cart pole moves very quickly. In the rest of the attacks/ unattacked videos, the cart poles do not move as fast as in illusory attack videos.". I think  this requires a closer investigation.

2. Unanswered question from my review -- "Also, why is "optimal illusory attack" defined? I do not see it being used in any part of the text."

Edit:
After the discussion phase, I have decided to increase my scores since the rebuttal cleared most of my questions.


**Summary Of The Paper:**

Summary:
This paper emphasizes developing statistically undetectable attacks for autonomous decision-making agents. The authors develop a novel class of illusory attacks that are consistent with environment dynamics. Their results show that illusory attacks can easily fool humans, unlike the previous attacks in the literature. They compare illusory attacks with other attacks and show their performance under different defense techniques.


Questions and comments:
1. In Table 2, what is the perturbation budget used for the attacks?
2. Based on Section 3, I think in definition 4.1 and the rest of the text, it should be P_{\varepsilon, \pi} and not P_{\pi, \varepsilon} for consistency.
3. In Section 3, \gamma is not defined.
4. In definition 4.3, it says the highest expected return but shows minimization in equation (2). Is this correct? If yes, why? Also, why is "optimal illusory attack" defined? I do not see it being used in any part of the text.
5. I would suggest using \tau \sim P_{\varepsilon, \pi} instead of \tau \sim (\varepsilon, \pi}) in equation (3).
6. In equations (3, 7), should the summand not be till T-1 and not T?
7. In equation (7), it should be \lambda instead of \gamma, right?
8. On page 8, it should be Table 1 and not Table 5.3 for consistency.
9. I think it's a better idea to replace Table 1 with Table 7.5.1 in the main text since the absolute reward values are easier to compare.

**Summary Of The Review:**

As mentioned in the weaknesses, I am not convinced if W-illusory attacks are undetectable. I think evaluations in more complex environments are required to strengthen the paper. The proposed attack is not robust to reality feedback. Hence, I do not feel that the contributions in this paper are very significant.

---

> ### Author Response · Authors · 2022-11-10
> **Part 2**
>
>
> A2.3: We discussed this, but find the current representation to be more interpretable. We are open to discuss this change further if it is seen as critical.
>
>
> > Strength And Weaknesses:
> > Strengths:
>
> >Focuses on developing attacks that are consistent with the dynamics of the environment.
> Develops a novel class of illusory attacks that are consistent with the dynamics. Introduces the concept of statistical indistinguishability for stochastic control processes.
> Well-structured sections. The supplementary material contains all the relevant videos related to human study.
> Weaknesses:
>
> >I think that the human study setup is biased. The samples given in the supplementary show that it's easy to detect attacked vs. unattacked videos. All the unattacked videos labeled in the study balance the pendulum or cart pole perfectly, while all the attacked videos do not do that. Hence, it should be easy to detect attacked vs. unattacked videos for these simple environments.
> >I think illusory vs. unattacked videos can also be figured out without much difficulty. For example, only in illusory attacks the cart pole moves very quickly. In the rest of the attacks/ unattacked videos, the cart poles do not move as fast as in illusory attack videos.
>
> A2.4: Thank you for this remark. Please note however that human participants were specifically asked to discriminate between the unattacked and attacked *environments*, not between different policies. We chose this question as a single environment often permits diverse (near-optimal) policies, hence distinguishing between policies is not generally a good predictor for adversarial attacks. Our study is *fair* since we asked participants the same question for our method and baseline attacks, resulting in significant differences.
> Further, note that none of the participants are experts in the field, and were left to their best judgement. A study with participants that are experts in RL constitutes an interesting direction for future work.
>
> >Given the above weaknesses, I am not sure why the detection accuracies are low for the illusory attacks in Table 1.
>
> A2.5: Please note that our detector does not test for statistical indistinguishability directly, as state-of-the-art methods to detect longer-term statistical correlations are very sample-inefficient [Shi et al, 2020], requiring impracticably large numbers of test-time samples for statistically-significant detection. In contrast to these works, our detector trades off between detection accuracy and sample efficiency. Please see section 7.3 for implementation details the detector which was used to generate results in Table 1.
>
> >What is the naive detection algorithm that is used for Table 1? Also, W-illusory attacks are not "statistically indistinguishable", unlike perfect illusory attacks, right?
>
> A2.6: This is correct. W-illusory attacks are a practical relaxation of illusory attacks. As stated in definition 4.5, “a W-illusory attack is an adversarial attack that is consistent with the victim’s model of the observation-transition probabilities mv”. […] “So importantly, W-illusory attacks can in general change the distribution of trajectories”.
>
> >I am not convinced if good W-illusory attacks should always exist. Pendulum and cart pole are very simple environments. I would suggest adding experiments with more complex environments. Not sure about transferability to other tasks.
>
> A2.7: Please note that definition 4.7 poses a practical implementation of W-illusory attacks. We are currently running experiments for W-illusory attacks on Mujoco, and will add results to our paper before the end of the weekend. Initial results can be retrieved fully anonymously here:
> https://drive.google.com/file/d/1ffNtT3RtvEQRAX3D9A2mezdappCVq-Ku/view?usp=sharing
>
> >Reality feedback is a practical and obvious way to defend against illusory attacks. Hence, the attack proposed in this paper does not appear to be strong.
>
> A2.8: Given suitable reality feedback, *any* observation-space adversarial attack can be mitigated, including MNP attacks. However, we are adamant to stress that, in the case of perfect illusory attacks, reality feedback can be the only way the only possible defence. We demonstrate how reality feedback can we used to robustify test-time policies. We’d like to reiterate that reality feedback is not always feasible.
> It’s a bit like saying that the answer to cybersecurity threats is to have perfectly secure channels.

---

> ### Author Response · Authors · 2022-11-10
> **Response Part 1**
>
> >Summary Of The Paper:
> >Summary: This paper emphasizes developing statistically undetectable attacks for autonomous decision-making agents. The authors develop a novel class of illusory attacks that are consistent with environment dynamics. Their results show that illusory attacks can easily fool humans, unlike the previous attacks in the literature. They compare illusory attacks with other attacks and show their performance under different defense techniques.
>
> >Questions and comments:
> >In Table 2, what is the perturbation budget used for the attacks?
>
> A2.1: The budget used here is 0.2, which is comparable to the budget used in the SA-MDP paper ([Zhang et al., 2020]).
>
> > Based on Section 3, I think in definition 4.1 and the rest of the text, it should be P_{\varepsilon, \pi} and not P_{\pi, \varepsilon} for consistency.
> In Section 3, \gamma is not defined.
> In definition 4.3, it says the highest expected return but shows minimization in equation (2). Is this correct? If yes, why? Also, why is "optimal illusory attack" defined? I do not see it being used in any part of the text.
> I would suggest using \tau \sim P_{\varepsilon, \pi} instead of \tau \sim (\varepsilon, \pi}) in equation (3).
> In equations (3, 7), should the summand not be till T-1 and not T?
> In equation (7), it should be \lambda instead of \gamma, right?
> On page 8, it should be Table 1 and not Table 5.3 for consistency.
>
> A2.2: Thank you for these remarks, we made fixes and added additional clarifications in the updated version of the paper.
>
>
> >I think it's a better idea to replace Table 1 with Table 7.5.1 in the main text since the absolute reward values are easier to compare.

---

> ### Author Response · Authors · 2022-11-17
> **Response to reviewer's response**
>
> >Response to rebuttal
>
> >Thanks for the rebuttal; however, some of my concerns remain. Could you please try to address these?
>
> >I am not convinced about illusory attacks not being detectable to "attentive human supervisors". I think a non-expert in RL can figure out attacked vs. unattacked in cases of illusory attacks. Quoting from my review -- "For example, only in illusory attacks the cart pole moves very quickly. In the rest of the attacks/ unattacked videos, the cart poles do not move as fast as in illusory attack videos.". I think this requires a closer investigation.
>
> We agree that the phrasing in section 5.4 was misleading; we have adjusted the writing. We now clarify that in our study, humans unfamiliar with adversarial attacks could not distinguish unattacked observation sequences from sequences attacked by W-illusory attacks. At the same time, we reiterate that W-illusory attacks are generally detectable (as also stated in Definition 4.5). Together, our study with humans thus emphasises the importance of bringing the existence of illusory attacks to the attention of safety practitioners, which could provide specialised training to human supervisors.
>
> >Unanswered question from my review -- "Also, why is "optimal illusory attack" defined? I do not see it being used in any part of the text."
>
> Thank you for the heads up. We have now cross-referenced more clearly, stating that W-illusory attacks are a relaxation of optimal illusory attacks.

---

> > ### Author Response · Authors · 2022-11-18
> > **score**
> >
> > Dear reviewer, given these clarifications, would you be willing to increase your score?

---

### Official Review · Reviewer_YTMM · 2022-11-02

**Confidence:** 4
**Correctness:** 2
**Technical Novelty And Significance:** 3
**Empirical Novelty And Significance:** 3
**Recommendation:** 5

**Clarity, Quality, Novelty And Reproducibility:**

Clarity: The paper is easy to follow. Algorithm some statements made in the paper are lack of rigorousness (explain in the previous section), the statements are clearly presented.

Quality: The problem formulation has some flaws as I mentioned in the previous section. I cannot discuss the quality of the algorithms and the experimental results before we address the problems in the formulation.

Novelty: Studying adversarial attacks with certain level of stealthiness is novel and interesting.

Reproducibility: I did not check the code and the experiment setup. I cannot provide judgement. I did not check the math thoroughly.

**Strength And Weaknesses:**

The problem studied in this paper novel and meaningful. That is to consider the stealthiness of adversarial attacks and the trade-off between stealthiness and effectiveness of the attacks. Stealthiness and detection are the important twins in security problems, especially security problems for sequential decision-making systems where attacks are launched not just one time but sequentially.

But the definition of indistinguishability and the detection mechanisms proposed throughout the paper are questionable. The assumptions made in Section 4 are contradicting with the RL algorithm that the author later choose as the focus of the study.

1. The definition of statistical indistinguishability is a strong one. If statistical indistinguishability holds, no detection algorithms can detect such attacks. But the attacks that can avoid being detected by some detection mechanisms don't necessary need to be statistically indistinguishable. The use of statistical indistinguishability as a condition to craft adversarial attacks can lead to no attacking strategy satisfying the condition. Consider an MDP with **continuous** state space. Can the authors give an example of an MDP and an attacking strategy $\pi_{adv}(s)\neq 0$ for some $s\in\mathcal{S}$ such that $\mathbb{P}_{\pi,\mathcal{E}} = \mathbb{P}_{\pi,\mathcal{E}'},\forall (\mathcal{S}\times\mathcal{A})^T$? To make the example simple, we can assume T=2. Such examples can help the readers better understand how strong the definition is.

2.   I agree with the authors that the assumption of the victim knowing a world model $m_v$ is not unrealistic. "victims can learn accurate world models from unperturbed train-time samples". However, if the victim knows the world model, what is the point of interacting with the world to observe the state? The victim can leverage a more efficient algorithm (value iteration, policy iteration + function approximation if the state space is large) instead of the algorithms discussed in the paper.

3. The detection mechanism used in the experiment is included in the paragraph starting with "Using a dynamics model to detect adversarial attacks." Please highlight the detection mechanism since it is an important factor for the experiments. If I understand it correctly, the detection mechanism considers observations in a single step and an attack is detected if the predicted observation and the actual observation is larger than a threshold $c$. Does this detection technique come from a reference? I have **three** concerns regarding the mechanism. 1. It does not consider the whole trajectory history to detect 2. how do you measure distance for discrete state space (how to set threshold c)? 3. if c is small, even if there is not attack, the actual observation can be different from the predicted observation given the stochasticity of the model.

Other minor comments:
* I came across a paper that studies adversarial attacks on rewards with a definition of stealthiness in it (C1). As a reader, I am curious about the difference between undetectability in attacks in reward and attacks in state observations?

* In optimal control or MDP or POMDP, adversarial attacks have been investigated by many researchers (C2, C3). These attacks are sometimes called false data injection attacks or sensor attacks. The detector of attacks is usually based on statistical evidence instead of single observations.

* Avoid using "may" in your statement. For example, in definition 4.1, a sampling policy that may conidtion on the whole history. The authors can say "a sampling policy where T can be infinity" or "a sampling policy that can condition on the whole action-observation history". In section 5 "may be unable to detect W-illusory attacks". Instead of using "may", scientific writing should specify under what conditions, humans are able to detect W-illusory attacks and under what conditions, human are not able to do so. Similar examples of using "may" can be found throughout the paper.


[C1] Huang, Yunhan, and Quanyan Zhu. "Deceptive reinforcement learning under adversarial manipulations on cost signals." International Conference on Decision and Game Theory for Security. Springer, Cham, 2019.

[C2] Mo, Yilin, and Bruno Sinopoli. "False data injection attacks in control systems." Preprints of the 1st workshop on Secure Control Systems. Vol. 1. 2010.

[C3] Pasqualetti, Fabio, Florian Dorfler, and Francesco Bullo. "Control-theoretic methods for cyberphysical security: Geometric principles for optimal cross-layer resilient control systems." IEEE Control Systems Magazine 35.1 (2015): 110-127.


**Summary Of The Paper:**

The paper studies adversarial attacks on the state observation (sensor inputs) channel of a RL agent. Different from previous work (in particular [Zhang et al., 2020]), the authors consider the stealthiness of the adversarial attacks. The contribution of the paper is that it defines the concept of detectability for adversarial attacks on state observation and proposes an algorithm that can compute and carry out such undetectable attacks.



**Summary Of The Review:**

The paper studies a novel and interesting problem of adversarial attacks on state observations in RL. The adversarial attacks are crafted in a way to avoid being detected. But the problem formulation has flaws. For example, the conditions posed for statistically indistinguishability, based on the reviewer's understanding, are very strong. The reviewer hopes to see some examples to support the definition. The assumption is not unrealistic itself. But it is unrealistic that the victim will use the algorithm discussed in the paper.  The detector discusses in the paper is not a detector one usually finds in cyber-security. It does not consider statistical evidence. There are other concerns regarding the detection mechanisms discussed in previous sections.

---

> ### Author Response · Authors · 2022-11-10
> **Part 2**
>
> A1.2: Thank you for this remark. In section 5.3. (Perfect Illusory attacks), we present results for the Pendulum and CartPole environments, which both have continuous state spaces. We will also add a 2-step example to the appendix.
>
> > I agree with the authors that the assumption of the victim knowing a world model mv is not unrealistic. "victims can learn accurate world models from unperturbed train-time samples". However, if the victim knows the world model, what is the point of interacting with the world to observe the state? The victim can leverage a more efficient algorithm (value iteration, policy iteration + function approximation if the state space is large) instead of the algorithms discussed in the paper.
>
> A1.3: The concepts we use in our paper can be used with both model-based (i.e. planning methods) or model-free approaches (which we explore in our paper), this is a good point and something to explore in future work.
> Just to be clear — having a world model which is used for planning does not get around the issue that the victim has to observe the current state (either as an input to the planning algorithm or to the amortised policy) and thus the observation can be attacked.
>
> >The detection mechanism used in the experiment is included in the paragraph starting with "Using a dynamics model to detect adversarial attacks." Please highlight the detection mechanism since it is an important factor for the experiments. If I understand it correctly, the detection mechanism considers observations in a single step and an attack is detected if the predicted observation and the actual observation is larger than a threshold c.
> >Does this detection technique come from a reference? I have three concerns regarding the mechanism. 1. It does not consider the whole trajectory history to detect 2. how do you measure distance for discrete state space (how to set threshold c)? 3. if c is small, even if there is not attack, the actual observation can be different from the predicted observation given the stochasticity of the model.
>
> A1.4: We agree that the detection mechanism used in our work does not guarantee statistical indistinguishability. However, despite its simplicity, it is highly effective at detecting state-of-the-art adversarial attacks. We similarly agree that tuning c, especially in stochastic environments, is essential to the detection mechanism. We tuned c such that no unattacked trajectories would be classified as attacked (please see section 7.3 for more details). In deterministic environments, other notions of state similarity can be adopted, such as nearest neighbour approaches.
> Furthermore, state-of-the-art methods to detect long-term statistical correlations are very sample-inefficient (Shi et al, 2020), requiring impracticably large numbers of test-time samples for statistically-significant detection. In contrast, our simple detector provides a trade-off between detection accuracy and sample efficiency.
>
> > Other minor comments:
> >I came across a paper that studies adversarial attacks on rewards with a definition of stealthiness in it (C1). As a reader, I am curious about the difference between undetectability in attacks in reward and attacks in state observations?
>
> A1.5: Thank you for pointing us to this related work in cybersecurity, we have added it to our related work section. The fundamental difference is that C1 assumes direct adversarial manipulation of cost/reward signals. We in contrast assume that the agent is not supplied with a reward signal at test time, as is usually the case during deployment.
>
> >In optimal control or MDP or POMDP, adversarial attacks have been investigated by many researchers (C2, C3). These attacks are sometimes called false data injection attacks or sensor attacks. The detector of attacks is usually based on statistical evidence instead of single observations.
>
> A1.6: Thank you for pointing out these works, we added them to the related work. C2 and C3 both study hard-coded sensor attacks on linear control systems and the detection mechanisms proposed are only applicable to such systems. In contrast, we present a novel reinforcement learning framework for end-to-end learnt attacks on high-dimensional non-linear systems.
>
> >Avoid using "may" in your statement. For example, in definition 4.1, a sampling policy that may conidtion on the whole history. The authors can say "a sampling policy where T can be infinity" or "a sampling policy that can condition on the whole action-observation history". In section 5 "may be unable to detect W-illusory attacks". Instead of using "may", scientific writing should specify under what conditions, humans are able to detect W-illusory attacks and under what conditions, human are not able to do so. Similar examples of using "may" can be found throughout the paper.
>
> A1.7: We improved our writing accordingly.

---

> ### Author Response · Authors · 2022-11-10
> **Response Part 1**
>
> > Summary Of The Paper:
> > The paper studies adversarial attacks on the state observation (sensor inputs) channel of a RL agent. Different from previous work (in particular [Zhang et al., 2020]), the authors consider the stealthiness of the adversarial attacks. The contribution of the paper is that it defines the concept of detectability for adversarial attacks on state observation and proposes an algorithm that can compute and carry out such undetectable attacks.
>
> >Strength And Weaknesses:
> >The problem studied in this paper novel and meaningful. That is to consider the stealthiness of adversarial attacks and the trade-off between stealthiness and effectiveness of the attacks. Stealthiness and detection are the important twins in security problems, especially security problems for sequential decision-making systems where attacks are launched not just one time but sequentially. But the definition of indistinguishability and the detection mechanisms proposed throughout the paper are questionable. The assumptions made in Section 4 are contradicting with the RL algorithm that the author later choose as the focus of the study. The definition of statistical indistinguishability is a strong one. If statistical indistinguishability holds, no detection algorithms can detect such attacks. But the attacks that can avoid being detected by some detection mechanisms don't necessary need to be statistically indistinguishable. The use of statistical indistinguishability as a condition to craft adversarial attacks can lead to no attacking strategy satisfying the condition.
>
> A1.1: While the reviewer’s statements are correct, we find that in many environments there are indeed perfectly indistinguishable attacks, which makes this a useful concept to introduce.
> Furthermore, we present full statistical indistinguishability as the extreme limit of our investigation, not as the central result. For example, we study W-illusory attacks, which are not necessarily fully statistically indistinguishable. In other words, statistical indistinguishability is a sufficient but not always necessary condition for illusory attacks.
>
> >Consider an MDP with continuous state space. Can the authors give an example of an MDP and an attacking strategy [...] such that $\mathbb{P}{\pi,\mathcal{E}} = \mathbb{P}{\pi,\mathcal{E}'},\forall (\mathcal{S}\times\mathcal{A})^T$? To make the example simple, we can assume T=2. Such examples can help the readers better understand how strong the definition is.

---

> ### Author Response · Authors · 2022-12-08
> **rebuttal**
>
> Dear reviewer,
>
> we were wondering if you were satisfied with the rebuttal provided and if you would be willing to increase your score?
>
> Thank you.
>
> The authors.

---

### Author Response · Authors · 2022-11-10
**General Response**

We thank all reviewers for their time and valuable feedback, and appreciate that reviewers found our work novel and relevant.

While three scores of 5 *suggest* consistent reviews, we find that each reviewer had their own concerns.
Some of these concerns were simply misunderstandings.
Some concerns were due to issues with the quality of the writing which we have now improved.
We now address each of the remaining individual concerns and we hope that the reviewers will take another open-minded look at the paper.

### Clarifications

We wish to highlight three central clarifications:
(1) In contrast to other works, we study **undetectable** attacks on **sequences** observations.
(2) Not all presented methods are perfectly undetectable. For example, the W-illusory attacks pose a trade/off between undetectability, adversarial objectives and computational efficiency.
(3) Reality feedback alters the problem statement and is effective against all adversarial attacks.

### Additional experiments
As suggested by the reviewers, we are **currently running experiments in MuJoCo**, and will add **more results over the weekend**.

**Initial video results for the Hopper and HalfCheetah** domains can be retrieved fully anonymously here:
https://drive.google.com/file/d/1ffNtT3RtvEQRAX3D9A2mezdappCVq-Ku/view?usp=sharing

Note that we ran the SA-MDP baseline with a perturbation budget slightly smaller than that in the original paper (Zhang et al, 2021).
Furthermore, we will add results for W-illusory attacks over the weekend.


### Revised paper
We have updated our paper to address the concerns raised. We highlighted changes in red (old) and blue (new).

We are more than happy to answer any follow-up questions.
Thank you.

The Authors

---

### Author Response · Authors · 2022-11-14
**Results for requested experiments**

Dear reviewers,

as suggested, we have run experiments in the MuJoCo environments Hopper and HalfCheetah and __added the core experiments' results to Table 1__ in the revised paper.
We found that in both environments, __100% of state-of-the-art SA-MDP attacks were detected, while 0% of the learned W-illusory attacks were detected__.
Please also consider the experimental video results that we provide anonymously at https://drive.google.com/file/d/1ffNtT3RtvEQRAX3D9A2mezdappCVq-Ku/view?usp=sharing.

We hope that these results clarify the reviewers' concerns about scalability. The results show that both *perfect illusory attacks* and *W-illusory attacks* scale to higher-dimensional environments, which are common RL benchmarks. We are more than happy to address additional questions.

Best wishes,
The authors

---

### Author Response · Authors · 2022-11-17
**Kindly asking for your response before the end of the revision period**

Dear reviewers,

As the revision period ends tomorrow, we would appreciate the reviewers confirming whether our response has addressed their concerns.
We want to stress again that we believe we were able to address and clarify all major concerns raised.

Thank you for your time.

The authors.

---

### Decision · Program_Chairs · 2023-01-20

**Decision:**

Reject

**Justification For Why Not Higher Score:**

See above for weaknesses.

**Justification For Why Not Lower Score:**

I recommend rejection.

**Metareview: Summary, Strengths And Weaknesses:**

The paper studies adversarial attacks on the state observation (sensor inputs) channel of an RL agent. Reviewers generally liked the problem studied in the paper of adversarial attacks on state observations in RL. However, they were not convinced by some of the arguments of the work regarding illusory attacks not being detectable to "attentive human supervisors". One reviewer argues the problem formulation has flaws as the conditions posed for statistically indistinguishability are very strong. I think the paper could benefit from having more experiments (i.e., additional environments), and that it could better explain the limitations of this work and the practicality of the proposed approach. Given all, I think the paper needs a bit more work before being accepted.